



**ERA5-based database of Atmospheric Rivers over Himalayas**
Munir Ahmad Nayak*, M. Farooq Azam, Rosa Vellosa Lyngwa
[1]Discipline of Civil Engineering, Indian Institute of Technology Indore, Simrol, Indore, Madhya Pradesh, India-453552
Correspondence to: Munir Ahmad Nayak, munir_nayak@iiti.ac.in





## 1 Abstract

Atmospheric Rivers (ARs) —long and narrow transient corridors of large horizontal moisture flux in the lower troposphere— are known to shape the hydrology of many regions around the globe. Heavy precipitation and flooding are often observed over many mountainous regions when the moisture-rich filaments impinge upon the elevated topographies. Although ARs and their impacts over many mountainous regions are well documented, their existence over the Himalayas and importance to the Himalayan hydrology have received negligible attention in the scientific literature. The Himalayas support more than a billion population in the Indian subcontinent, sustain the region's biodiversity, and play important roles in regulating the global climate.

In this study, we develop a comprehensive database of ARs over the Himalayas using the European Reanalysis fifth generation (ERA5) fields of humidity and winds. The AR database consists of the dates and times of ARs from 1982 to 2018, their duration, major axes, and intensities and categories. We find that majority of intense ARs are associated with extreme precipitation widespread over the Ganga and Indus basins of the Himalayas, suggesting that ARs have profound impacts on the hydrology of the region. The AR database developed here is envisioned to help in exploring the impacts of ARs on the hydrology and ecology of the Himalayas. For this, we provide a few brief future perspectives on AR-Himalayas relationships.

The data developed in this study has been uploaded to the Zenodo repository at https://doi.org/10.5281/zenodo.4451901 (Nayak et al., 2021). The data are also included in the Supplemental Information for easier access.





## 2 Introduction

### 2.1 Brief overview of Atmospheric River studies over mountainous regions

Owing to the uneven solar heating of the earth's surface, net transports of atmospheric energy and moisture result from the tropics to the polar regions. Newell et al., (1992) observed driver-like filaments in the maps of specific humidity and horizontal water vapor flux in the lower troposphere, which they named as "tropospheric rivers". The features were later termed as "Atmospheric Rivers" (ARs) by Zhu & Newell, (1998), who estimated that more than 90% of the poleward flux of water vapor in the extratropical atmosphere at any time instant happens through only a few transient (less than ten) ARs that occupy ~10% of the global circumference at a given latitude. Two decades later, after significant research breakthroughs in AR science, a more formal and thorough definition was introduced in the Glossary of Meteorology (Ralph et al., 2018). The most definitive feature of ARs is the strong horizontal water vapor transport in long and relatively narrow bands in the lower atmospheric levels (< 3km from the surface of Earth). ARs are typically associated with extratropical cyclones, many ARs, however, transport moisture from the tropical latitudes and are associated with tropical cyclones (Cordeira et al., 2013; Ralph et al., 2011; Sodemann and Stohl, 2013; Stohl et al., 2008).

In the last two decades, ARs have gained increased scientific attention due to their societal impacts in the form of intense storms and flooding events associated with them. Over many regions around the globe, ARs are known to be among the key drivers of major flooding events and other extreme precipitation related disasters (west coast of USA (Dettinger et al., 2011; Dettinger and Cayan, 2014; Young et al., 2017); Europe (Lavers et al., 2012; Lavers and Villarini, 2013b, 2015a); Japan (Kamae et al., 2017)). At the same time, moisture transported in ARs has been shown to be a major source of beneficial water to sustain municipal water systems, agricultural systems, and the ecosystem (Florsheim and Dettinger, 2015; Guan et al., 2010), and at times, they supply essential water to recover from droughts (Dettinger et al., 2011). In US West Coast mountainous regions, a major fraction (more





than 30%) of winter snowpack is shown to be accumulated by only a few AR storms (Guan et al.,
2010; Guan and Chan, 2006).
In mountainous ranges, the impacts of ARs are pronounced when the moisture-laden air is
lifted vertically by the elevated topography, such is the case with the US West Coast, parts of Europe,
and many other regions. In the Indian subcontinent, an important barrier to atmospheric flow is the
Himalayan mountain range; however, the role of ARs in the hydrology of the region is essentially
unexplored.
2.2    Himalayan water resources under climate change
The Himalayan mountain range is home to the world's highest glaciers therefore rightly called
the water tower of Asia or 'Third Pole' (Bolch et al., 2019). The Himalayan river basins of Indus,
Ganga, and Brahmaputra are among the most melt-water dependent rivers systems on Earth and satisfy
the water requirements of more than a billion people (Immerzeel et al., 2020; Pritchard, 2019). There
are ~40,000 glaciers covering ~41,000 km$^2$ area in the Himalayas with an estimated ice volume of
3500 km$^3$ (Bolch et al., 2019; Farinotti et al., 2019). The frozen water storage in these glaciers and
seasonal snow covers directly affect the runoff in the Himalayan river basins hence is important to the
downstream population for irrigation, hydropower, sanitation, and industries (Biemans et al., 2019;
Bolch et al., 2019; Momblanch et al., 2019).
Under the ongoing global warming, water towers of the world's mountain ranges are under
threat including the Himalayas —Indus Basin being the most vulnerable (Huss and Hock, 2018;
Immerzeel et al., 2020). Glacio-hydro-meteorological investigations in the Himalayas are scarce
compared to other mountainous regions on Earth because of inherent challenges in maintaining the
monitoring networks due to rough climatic conditions and challenging access to the glaciers (Azam et
al., 2018). A few studies, generally using low altitude meteorological data, revealed increasing
temperature trends over the past 5–6 decades with varying rates in different regions whereas



precipitation did not show any particular spatiotemporal trend in the Himalayan region (Krishnan et
al., 2019).
In response to the global as well as regional warmings (Banerjee and Azam, 2016; Krishnan et
al., 2019; Pörtner et al., 2019), glaciers have been found to be diminishing in the Himalayas over the
past 5–6 decades (Azam et al., 2018; Bolch et al., 2019). However, the shrinkage rates are
heterogeneous: negative mass balances in the eastern, central, and western Himalaya and near-balance
state in the Karakoram range (Azam et al., 2018; Bolch et al., 2019; Brun et al., 2017). These changes
caused considerable alteration to the volumes and seasonality in river runoff, affecting the agriculture,
hydropower production, and even sea level (Biemans et al., 2019; Farinotti et al., 2019; Momblanch
et al., 2019). For instance, river flow reduction by 1% is expected to bring roughly 3% reduction in
hydropower production (Laghari, 2013). Glacier changes are also linked with potential hazards such
as glacial lake outburst floods (Bhambri et al., 2020; Veh et al., 2019). Besides, the Himalayan rivers
are trans-boundary in nature and any future change in runoff regimes may give rise to demands for
reworking water-sharing treaties among surrounding countries (Molden et al., 2017)
2.3      Lack of studies on ARs in Himalayas
Over the years, especially since the availability of satellite-based imagery, ARs have garnered
numerous important works that have significantly advanced the scientific understanding of the hydro-
climatology of many mountainous regions on Earth. For example, after understanding and recognizing
the presence and impacts of ARs, it has now been possible to more clearly define and narrow the
challenges in water management of the western US (Ralph et al., 2019; Steinschneider et al., 2015;
Steinschneider and Brown, 2012). Many previously unexplored mountainous regions begin to
appreciate the role of ARs in shaping the regional hydrology (Chen et al., 2019; Wille et al., 2019).
Knowledge of AR presence over the Himalayan mountain range and their hydrological impacts
is sparse. In some parts of the India continent, however, a few studies on ARs have emerged recently.
Lakshmi & Satyanarayana, (2019) observed that the heavy precipitation events that led to major



flooding in November and December of 2015 over Chennai city in south-east India were associated
with ARs. Thapa et al., (2018) studied the association of extreme precipitation over the western Nepal
with ARs penetrating the transect at 80.25°E from 27°N to 30.75°N, which intersects the Himalayan
arc used in this study only at one location. They highlighted that a significant fraction (more than 35%)
of annual maximum and seasonal maximum precipitation events happened during the presence of
ARs. Liang & Yong, (2020) used horizontal integrated water vapor transport (IVT) threshold of
$500\ kgm^{-1}s^{-1}$ to identify ARs and discussed their impacts over the Asian Monsoon regions, which
also covers the Indian subcontinent. However, the Himalayas, especially, the western Himalaya, are
less affected by the main strong monsoon flows, and IVT of $500 kgm^{-1}s^{-1}$ or higher rarely happens
over the region (as shown in Figure 2 of Liang & Yong, (2020)). Consequently, the study did not
identify ARs over the Himalayas. Another study over the east and west coasts of India by Laskhmi &
Satyanarayana, (2020) discussed AR climatology and their impact on extreme precipitation. In these
studies, significant fraction (up to 40%) of extreme precipitation was observed to be associated with
ARs.

Above studies mainly focused on the southern India and did not identify ARs over the

Himalayas. However, as noted earlier, orographic barriers such as the Himalayas are the most
favorable locations where ARs exert pronounced impacts due to strong potential for vertical uplifting
of moisture (Hughes et al., 2014; Neiman et al., 2013; Rutz et al., 2014). In addition, a large population
(more than 1 billion) live in the Himalayan basins and are heavily dependent on glacier- and snow-fed
river systems for their municipal, hydropower, and agricultural water supplies.
2.4        Objectives

The objective of this paper is to develop a dataset of ARs that penetrate the Himalayan transect.

The motivation of this work lies in the hypothesis that ARs exist and provide important contributions
to the water resources of the Himalayan basins. Though this has not been realized in the literature, we



believe that an easily accessible and freely available AR database can invigorate research on ARs and
the Himalayan hydrology and will serve to advance the science of global ARs.

## 3    Study Area and Climate

The Himalayan mountain range, west to east extent, is ~2700 km long and has three major
river basins i.e., Indus, Ganga, and Brahmaputra (Figure 1). These river basins cover an area of 2.75
million km$^2$ and contain some of the highly irrigated areas of the world with a total irrigated area of
577000 km$^2$ and the installed hydropower capacity of ~26000 MW.
Due to high altitudes and geographical location, the Himalayas together with Tibetan Plateau
provides a physical barrier that plays key role in global weather patterns by acting as a heat source
during the summer and a heat sink during the winter (Dimri et al., 2015). Climate of the Himalayas is
complex because of the impact of two major circulation systems i.e. Indian summer monsoon and
Western disturbances (Schiemann et al., 2008; Webster et al., 1998) (Figure 1). The Indian summer
monsoon and WDs are associated with the movement of Intertropical Convergence Zone (ITCZ),
which develops because of seasonal temperature and pressure differences between the northern and
southern hemispheres (Gadgil and Joseph, 2003). During the summer season, Tibetan Plateau heats
up and creates a void, to fill this void ISM originates from Indian Ocean, Arabian Sea and Bay of
Bengal and move northward providing the maximum precipitation during June-September months
(Maussion et al., 2014). Conversely, during winter season, Western Disturbances originate in the
Mediterranean or West Atlantic region and travel across Iran, Iraq, Afghanistan, Pakistan and India
and provide precipitation over the Himalaya (Rao et al., 1969).
The Indian summer monsoon intensity decreases from east to west while WDs weaken from
west to east along the Himalayan range (Bookhagen and Burbank, 2006, 2010; Maussion et al., 2014).
Besides, there are large differences in precipitation amounts across south-north axis of this mountain
range because of its orography (Azam et al., 2014b; Maussion et al., 2014). The Himalayas act as a
barrier to the monsoon winds, causing huge orographic precipitation on the south slopes with a south-
north gradient in the monsoon intensity (Dimri et al., 2015). Due to geographical location and regional
orography, very wet regions coexist with very dry regions.

## 4   Data and Methods

### 4.1      Data

We use column integrated water vapor transport (IVT) to identify ARs, for which the required
inputs are specific humidity and horizontal wind fields at multiple vertical levels of the troposphere.
These fields are extracted from ECMWF's fifth generation reanalysis ERA5 (Hersbach et al., 2020)
at 6-hourly temporal resolution, i.e., for each day, the fields are retrieved at four timesteps 00UTC,
06UTC, 12UTC, and 18UTC. ERA5 is improved with respect to its predecessor reanalysis data ERA-
Interim through assimilating additional observational data and adopting new developments in physical
modeling and data assimilation algorithms in its integrated forecasting system model. Significant
improvements have been observed in simulating the atmospheric dynamics of the troposphere, which
is where ARs are typically located; refer Hersbach et al., (2020) for details on ERA5 development.
The data spans from the year 1979 to the present and covers the entire globe at a high horizontal
resolution of 0.25°×0.25 (roughly 31km grid cell) at 137 vertical levels from the surface of earth.
The AR identification algorithm used in this study, discussed in next section, can at times
confuse large-scale cyclones as ARs, and hence, these need to be removed before labeling the
identified features as ARs. The dates and times of cyclones over the Bay of Bengal, Arabian Sea, and
continental India are retrieved from the cyclone database maintained by the Regional Specialized
Meteorological Centre (RSMC, www.rsmcnewdelhi.imd.gov.in), New Delhi, for tropical cyclones
over the North India Ocean. The southern face of the Himalayas receive large amounts of orographic
precipitation (Dimri et al., 2015; Maussion et al., 2014), which varies spatially and seasonally.
Unfortunately, no satellite or reanalysis product is capable of capturing spatial variation and magnitude
of precipitation over mountainous regions of the Himalayas (Andermann et al., 2012; Immerzeel et



al., 2015; Shen and Poulsen, 2018). Hence in this study, we use observation-based precipitation data
provided by the India Metrological Department (IMD, Pai et al., (2014), www.imd.gov.in). Since large
storms are normally expected during intense ARs, the concurrence of AR days with heavy
precipitation from IMD will, to some extent, provide an independent verification of ARs identified in
ERA5. The precipitation data is available at daily temporal and 0.25°×0.25° horizontal resolutions.
4.2      AR Identification
ARs are characterized by large moisture anomalies in the atmosphere, and since the availability

of water vapor imagery from satellites, many studies used column integrated water vapor (IWV)
observations from Special Sensor Microwave Imager/Sounder (SSM/I) to identify ARs (Neiman et
al., 2008a, 2013; Ralph et al., 2004; Zhu and Newell, 1998).

Later, however, it was realized that IWV does not incorporate the flux component of ARs, and

the column integrated water vapor transported (IVT), which includes horizontal winds in its
computation, is taken as the preferred metric to identify ARs in most of the recent studies. IVT offers
certain advantages over IWV in more coherently identifying ARs that show better relationship with
precipitation and are more skillfully predicted by numerical weather prediction models (Cordeira et
al., 2017; Dettinger et al., 2018; Nayak et al., 2014; Waliser and Guan, 2017; Young et al., 2017). IVT
magnitude on a grid cell is computed as the magnitude of column integrated zonal and meridional
moisture fluxes, as given in Equation 1 below.
$$IVT = g^{-1} \sqrt{\left( \int_{1000hPa}^{300hpa} qudp \right)^2 + \left( \int_{1000hPa}^{300hpa} qvdp \right)^2}$$

(1)

where, $IVT$ is magnitude of water vapor transport in $kg.m^{-1}.s^{-1}$, $q$ is specific humidity in $kg.kg^{-1}$,
$u$ and $v$ are, respectively, zonal and meridional wind velocities in $m.s^{-1}$, and $g$ is the acceleration due
to gravity ($9.81ms^{-2}$).



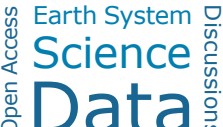

Many algorithms are available to identify ARs using IVT fields (Lora et al., 2020; Shields et
al., 2018); here, we use the algorithm developed by Lavers et al., (2012) for its conceptual and
computational simplicity. The algorithm has been successfully employed in many AR studies over the
US West Coast (Barth et al., 2017), Europe (Lavers &Villarini, 2015a, 2015b), and the central US
(Lavers &Villarini, 2013a; Nayak et al., 2016; Nayak & Villarini, 2017).
[Step 1] The first step of the algorithm is to draw a geographical transect for the study region,
and the ARs that penetrate it are selected as the ARs for the study area. For our study region, a transect
based on ERA5 grid cells was selected at the southern front of the Himalayas as shown in Figure 1a.
The transect (~1000 m a.s.l.) covers the entire arc of the Himalayas, except the elevated topography
ahead and below the eastern and western Himalaya, respectively, since these will significantly reduce
the lower-level moisture flow, which is a typical characteristic of ARs.
[Step 2] The next step is to find a threshold for IVT, above which an IVT field for any time
instant may be considered a potential AR instant. As in many studies, the 85th percentile of daily IVT
distribution is considered as the threshold. At each grid cell on the transect (Figure 1a), we take the
85th percentile of the daily IVT distributions, *i.e.*, 40 values for each day for the period 1979–2018.
The daily thresholds are smoothed by taking a 15-day moving average centered on that day. Owing to
latitudinal atmospheric temperature variations, the mean atmospheric water vapor and its flux varies
latitudinally, decreasing northward, hence we considered latitudinally-varying thresholds. As can be
noted, however, the transect is formed of numerous grid cells; thus, for simplicity, we divide the
transect in five bins (from BinA for the eastern Himalaya to BinE for the western Himalaya), as shown
in Figure 1a. For each bin, threshold for each day of year is taken as the average of the thresholds of
the grid cells forming the bin. This way we construct daily- and latitudinally-varying IVT threshold
series (Figure 1b).
[Step 3] The next step involves searching for a potential AR major axis from the IVT field at
a given timestep. For the timestep, the maximum IVT grid cell among the grid cells of the entire



transect is identified, and its IVT is compared with the threshold of the bin where the grid cell is
located. This maximum IVT grid cell is taken as the first point of major axis for the potential AR
timestep. As in Barth et al., (2017) taking the grid cell as the reference location, three grid cells on the
next longitude eastwards are searched for the next point of the potential major axis. If the maximum
IVT among the three grid cells is higher than the threshold, the new grid cell, where the IVT is highest,
is saved as the second point of the major axis, and the search proceeds eastward from it. The search
proceeds until either the IVT falls below the threshold or the length of the AR major axis is larger
than $2000\ km$, at which point the timestep is considered as an AR timestep. Most studies take
$2000\ km$ as the length criterion for identifying ARs (Fish et al., 2019; Guan and Waliser, 2015;
Neiman et al., 2008b; Wick et al., 2013; Zhou et al., 2018) though smaller length criterion of 1500 km
are also taken in some studies (Lakshmi and Satyanarayana, 2019; Lavers et al., 2013; Lavers and
Villarini, 2013b; Ramos et al., 2016; Thapa et al., 2018). Similarly, a southward search is made, and
the dates and time instances of all AR time steps identified are stored along with IVT values at grid
cells of the major axes.
[Step 4] Tracking only along the longitudes (i.e., by searching only southward) and then only
along the latitudes (*i.e.*, by searching only southward) allows us to identify only long rivers and remove
most of the small-scale cyclonic structures. We also remove cyclones in RMSC database that are
misidentified as potential AR timesteps. This is done by excluding all 6-hourly times of all cyclones,
from genesis to termination, from the timesteps earlier identified as AR timesteps. Though ARs are
often associated with cyclones, they are distinct from cyclones (Guan and Waliser, 2019; Pan and Lu,
2019), and it was deemed necessary to remove all cyclones regardless of the basin of origin, since all
basins considered in RMSC database are in close proximity to the study region.
[Step 5] After removing cyclones from AR timesteps, 18-hour persistence criterion was
adopted to exclude short-lived AR structures. From here onwards, we define an AR or AR-event as
the one when AR conditions are met for a minimum of three consecutive six-hour timesteps (18-





hours), i.e., we only consider persistent ARs. The time for which AR conditions sustain is taken as the
duration of the AR. The 18-hour duration criterion is implemented in most studies on hydrologic
impacts of AR (Albano et al., 2020; Lakshmi and Satyanarayana, 2019), since ARs usually cause small
magnitude precipitation in the initial 18 hours (Ralph et al., 2019; Rutz et al., 2014)

The criteria of one-way searches, excluding cyclones, and 18-hour persistence are important,

since many large-diameter IVT "blobs", which would have otherwise been confused as ARs, are
removed. Over the eastmost Bin, BinE, a few ARs identified appear like "blobs" (not shown here)
may be due to elevated topography just south and east of the bin (Figure 1a), which diverts the ARs
from their expected trajectory towards the Himalayas. These are considered as ARs here, though it
might be more reasonable to exclude them from AR database.

[Step 6] Now, it is likely that many ARs identified in eastward search also appear in southward

search. In order to avoid double counting these ARs and to maintain their spatial and temporally
coherency, all ARs were merged to form a single AR database as follows. For ARs in which at least
one timestep overlapped in both searches, the AR for which average IVT magnitude (averaged taken
over the entire length and duration of AR) was greater was saved in the AR database, and the smaller-
IVT AR was discarded. ARs that did not overlap in the searches were concatenated to the database
without any change. The developed AR database in this study consists of dates and times of ARs, their
major axes, duration, and their intensities and categories as defined in the next section.
4.3        AR Categorization

As seen in the literature, ARs produce diverse range of hydrometeorological impacts over

many regions of the globe; while many ARs supply beneficial water, some generate disastrous
precipitation. In an attempt to distinguish between beneficial and hazardous ARs for operational
purposes of weather prediction centers and water managers, Ralph et al., (2019) suggested a scale to
characterize ARs. The two most important factors that govern the strength and impact of an AR are
its IVT intensity and duration. It is noted that although IVT magnitude signifies an AR's impact in



terms of rainfall intensities it generates, the duration mainly governs the storm-total precipitation and
runoff variability (Lamjiri et al., 2017; Nayak et al., 2016; Ralph et al., 2013). Based on these
considerations, Ralph et al., (2019) proposed AR categories (Cats) in their Table 2 and Figure 4, where
ARs falling in Cat 1 to 2 are mostly beneficial, and those in 3 to 5 are hazardous, with 5 recognized as
the most hazardous. Ralph et al., (2019) mainly focused on ARs landfalling over the US West Coast,
where the IVT threshold to define an AR is 250 $kgm^{-1}s^{-1}$, and thus Cat 1 AR IVT intensity ranges
from 250 $kgm^{-1}s^{-1}$ to 500 $kgm^{-1}s^{-1}$, Cat 2 ranges between 500 $kgm^{-1}s^{-1}$ to 750 $kgm^{-1}s^{-1}$,
Cat 3 ranges between 750 $kgm^{-1}s^{-1}$ to 1000 $kgm^{-1}s^{-1}$, Cat 4 ranges between
1000 $kgm^{-1}s^{-1}$ to 1250 $kgm^{-1}s^{-1}$ and Cat 5 ranges from 1250 $kgm^{-1}s^{-1}$ onwards.

However, we noted that the climate of the Himalayas is different, and as will be shown later,

the threshold of IVT varies significantly with seasons, hence some region-specific modifications to
the AR categorization are necessary. AR categories adopted here are presented in Table 1, where AR
intensity is given by $IVT_{\max}$, which is defined as the maximum IVT at the first grid cells of the major
axes of the AR timesteps, *i.e.*, maximum over the starting IVTs where the search begins in each
timestep of the AR. This definition of AR intensity is consistent with that given in Ralph et al., 2019.
The major difference in AR categorization here and in Ralph et al., (2019) is that, unlike here, the
events in the first row of Table 1, where AR intensity is less than $250 kgm^{-1}s^{-1}$ (i.e., from threshold
value to $250 kgm^{-1}s^{-1}$), are not considered as ARs in Ralph et al., (2019). Indeed, this categorization
is not definitive and future research will elucidate the strengths and impacts of ARs over the
Himalayas, and it will help in developing more precise operational categorization. As a starting point,
however, the categorization in Table 1 can be taken as basis for understanding the climatology and
impacts of weak and strong ARs.

Table 1: AR categorization based on AR Intensity ($IVT_{max}$) and duration. Six AR categories are defined, with
categories Cat 0 to Cat 2 taken as primarily beneficial and Cat 3 to Cat 5 are taken as primarily hazardous, as
in Ralph et al., (2019).





| $IVT_{max}$ $(kgm^{-1}s^{-1})$ | AR Duration (hours) | | |
|---|---|---|---|
| | $<= 24$ | $24 - 48$ | $> 48$ |
| $\leq 250$ | 0 | 1 | 2 |
| $> 250 - 500$ | 1 | 2 | 3 |
| $> 500 - 750$ | 2 | 3 | 4 |
| $> 750 - 1000$ | 3 | 4 | 5 |
| $> 1000 - 1250$ | 4 | 5 | 5 |
| $> 1250$ | 5 | 5 | 5 |


4.4      AR tracks

The database developed in this study also provides the major axes of ARs, which can be loosely

taken as the trajectories or tracks of ARs. The tracks of ARs provide critical information regarding
their origin, moisture sources, and seasonal variability. As defined earlier (section 4.2), the major axis
of an AR timestep is the collection of grid cells over which IVT is maximum. ARs are not stationary;
the AR major axis changes from one timestep to the next. We collect all the major axes of an AR event
and define the average of those axes as the track of the AR. Most AR timesteps have lengths more
than the required $2000km$; here we show average over the length of the shortest major axis. For each
season, we show tracks of 25 random ARs to give an overall picture of their likely trajectories in
different seasons. The detailed analyses of moisture sources and trajectories of ARs is out of scope of
the present study and is left for future studies.
**5     Results and Discussion**

5.1      AR events, frequency and trends

IVT threshold varies significantly with seasons and across latitudes (Figure 1b). The lowest

values are observed in winter, following which the threshold rises abruptly in the Ganga Basin (within





30 days the threshold increased from $200 kgm^{-1}s^{-1}$ $to$ $500 kgm^{-1}s^{-1}$ in the southernmost bin, Bin
E, peaking in summer and then again dropping towards the end of autumn (Figure 1b). The peak IVT
in summer can likely be attributed to the Indian summer monsoon circulation, though it can be noticed
that seasonal variation is less pronounced in the Indus Basin transects, bins A and B, which suggests
that the Indus Basin is not heavily influenced by the monsoon. The monsoon reach and intensity over
the Himalayas is also reflected in the large differences of summer thresholds across the bins; the
smaller differences in winter, on the other hand, may be due to small south-north variations of air
temperature and atmospheric moisture. Unlike a fixed threshold of $200 kgm^{-1}s^{-1}$ used in most AR
studies (Hecht and Cordeira, 2017; Lavers et al., 2012; Leung and Qian, 2009) the seasonally-varying
threshold used here is aimed at extracting ARs from the background monsoon, due to which the normal
IVT over the Himalayas is significantly higher than $200 kgm^{-1}s^{-1}$ in summer (Figure 1b).

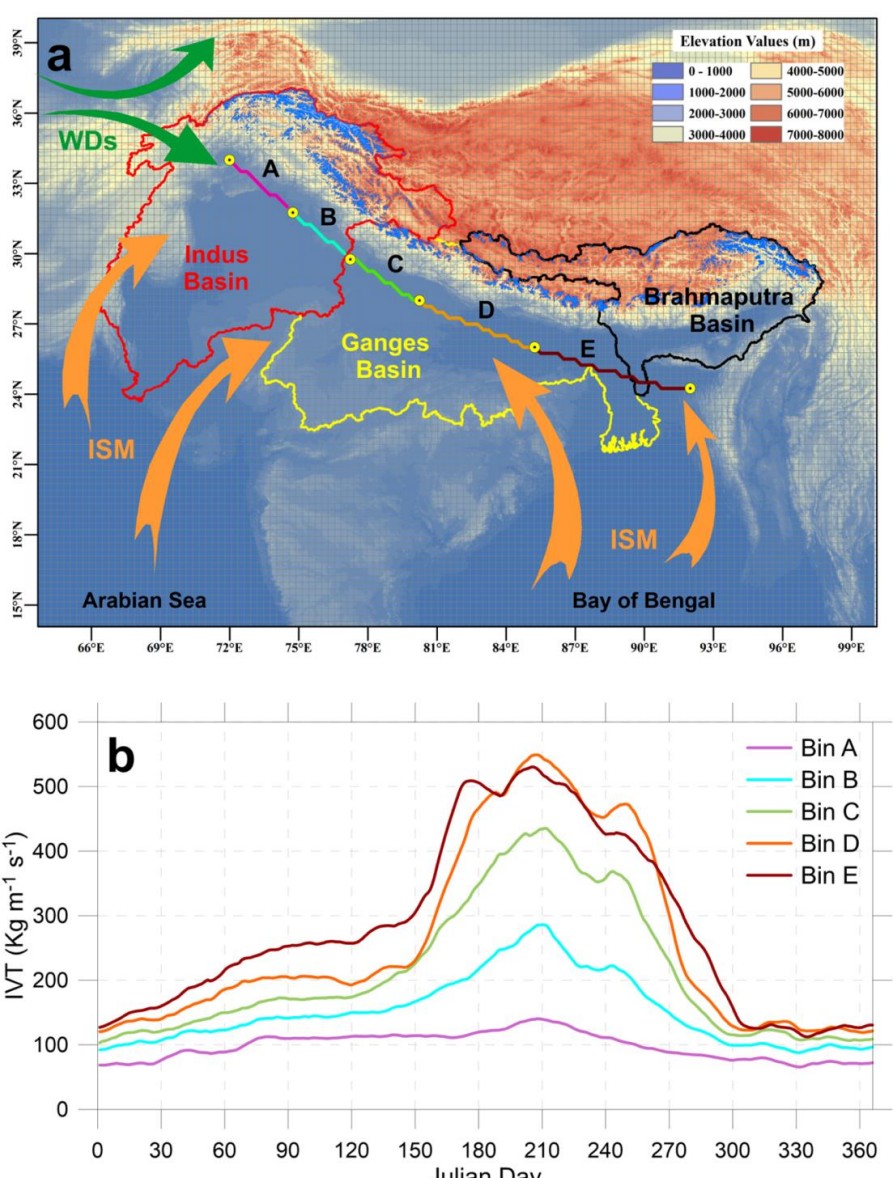

**Figure 1:** Study Area and Climate. Panel "a" shows the location of the selected transect (Bins A-E) along the
Himalayan front (~1000 m.a.s.l.). Orange and green arrows show the major circulations affecting the region,
the Indian summer monsoon and the western disturbances, respectively. The dark blue color in Indus, Ganges
and Brahmaputra basins shows the glacierized area from GAMDAM inventory *(Nuimura et al., 2015)*.
Background gridlines show the ERA5 horizontal grid spacing. Panel "b" shows the daily-varying 85th
percentile of IVT distribution along the bins. See Data and Methods section for more details.

The frequency of ARs varies annually and marginally across seasons, with annual average of

23 and seasonal averages of 7, 5, 6, and 5 ARs in winter (December–February), spring (March–May),
summer (June–August), and autumn (September–November), respectively (Figure 2a). The minimum
ARs of 15 were found in 1987 while the maximum of 37 in 2006. The year-to-year variability in
annual frequency marginally displays an oscillation of period roughly 3 to 4 years, which may be
related to large scale climatic patterns (Collow et al., 2020). It can be noticed that over the 1990s, the
frequency of ARs was consistently higher than the average frequency (Figure 2a).

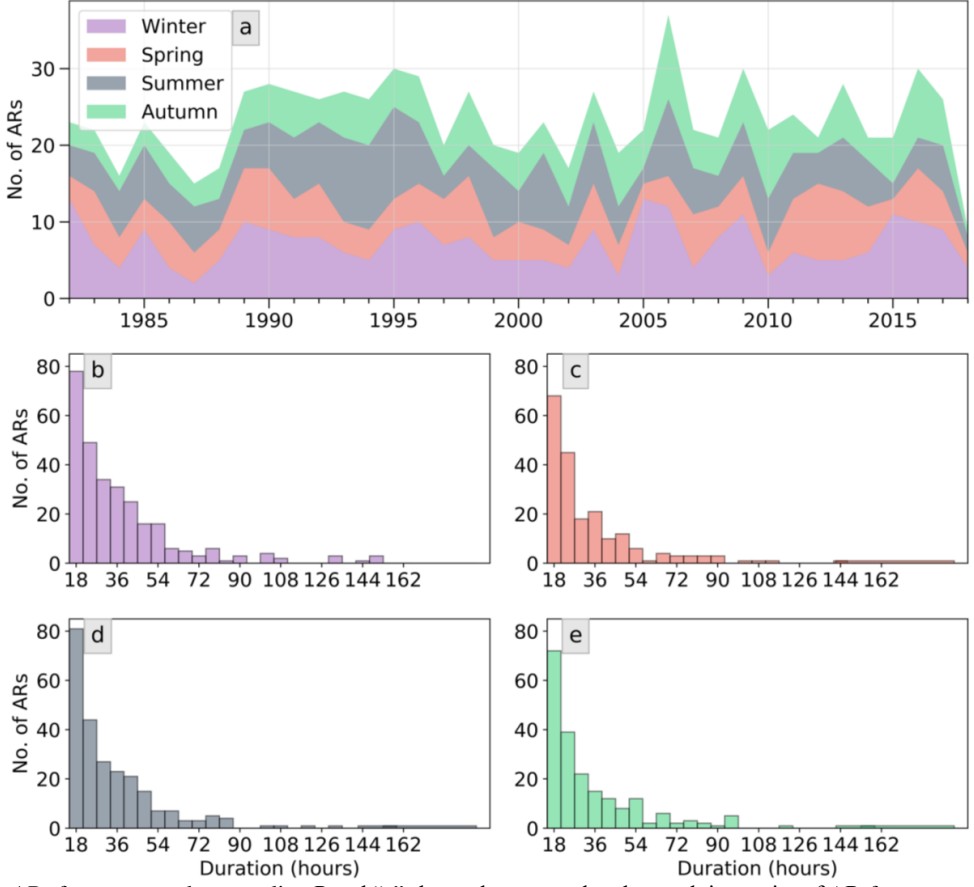

**Figure 2:** ARs frequency and seasonality: Panel "a" shows the seasonal and annual timeseries of AR-frequency
over the Himalayas. Panels "b–d" show the distribution of duration of ARs in different seasons.

For discovering any temporal trends, the AR frequency of the four seasons is regressed on time

(year) as an independent variable. The regression analysis yielded slopes close to zero for winter and
spring seasons ($p$ −values for slopes close to 0.8, which implies that the null hypothesis of no temporal
trends cannot be rejected; see Figure S1 in Supplemental Material). However, a decreasing trend (-0.5



ARs per decade, $p - value \cong 0.25$) and an increasing trend (+0.7 ARs per decade, $p - value \cong$
0.04, i.e., significant at 5% level) in summer and autumn frequencies of ARs are observed
respectively. The decline in summer ARs may be linked to the weakening of summer monsoon
highlighted in some recent studies (Mishra et al., 2012; Paul et al., 2016). This, however, would
suggest that observed weakening of summer monsoon maybe in fact weakening of summer ARs, and
the monsoon itself may not be weakening. More in-depth studies on trends in AR frequency of autumn
and summer, and its links with the monsoon weakening may bring some interesting insights into light.
The duration of most ARs (more than 90%) is less than three days (Figure 2b-e); however,
long-duration ARs that last for more than 72 hours (3 days), are also observed in all seasons. Winter
ARs are in general longer lasting (median duration 30 hours) than those in other seasons, for which
the median duration 24 hours.
Figure 3 shows the seasonal and category-wise distribution of ARs across the transect. It can
be seen that winter is the most prominent season for AR occurrences across the Himalayas, except for
the eastern Himalaya BinE (Figure 3). The higher frequency in winter can be related to increased
meridional tropospheric temperature gradient, where baroclinic instability increases the likelihood of
cyclone formation and WDs (Hunt et al., 2018b; Rao et al., 1971). Only a few ARs are observed over
BinA in summer, which again suggests that the Indian Summer Monsoon has little effect on the climate
of the Indus Basin. In contrast, the southernmost transect BinE experiences the highest frequency of
ARs in summer, which suggests that most ARs over the eastern Ganga Basin are linked to the Indian
Summer Monsoon. Except for extreme Bins (BinA and BinE), all the other bins have nearly uniform
frequency of ARs in spring, summer and autumn (Figure 3).

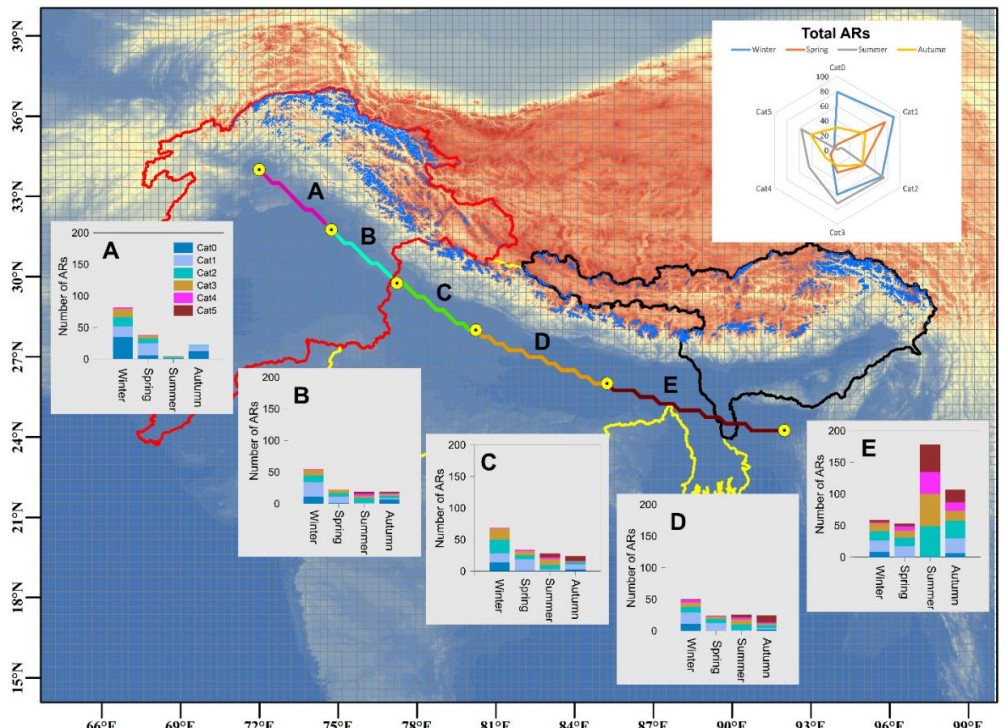

**Figure 3:** ARs Categories: Panel "A–E" show the distribution of ARs categories in different seasons over the five bins (A to E) along the transect. The inset shows the fractional distribution of AR categories in different seasons.

Most ARs that penetrate the Indus Basin are of weak categories 0 and 1, with only a few falling in disaster categories Cat 4 and 5; in fact, no Cat 5 AR is observed in western Indus BinA. For the eastern Ganga and Brahmaputra basins (BinE), majority of ARs are of Cat 3 or higher, since most of them happened in the monsoon season when the moisture transport in the atmosphere is in general large (Bookhagen and Burbank, 2006; Maussion et al., 2014). These higher category ARs over Bin E are most probably the reason for the frequent flooding in Nepal, NE India, and Bangladesh (Thapa et al., 2018; Yang et al., 2018). The radar diagram (inset in Figure 3) suggests that a large faction of ARs in summer are Cat 3 and higher, where for other seasons, majority of ARs are of Cat 3 or lower, i.e., mostly beneficial.



## 5.2    AR tracks

The major axes of 25 random ARs in each season, and each axis is averaged over the AR's life span, are plotted as lines in Figure 4; these lines thus represent the typical tracks of ARs in different seasons. From these tracks, we observe that majority of the ARs in winter and spring travel (the word *travel* is used loosely here to indicate axis of AR moisture transport) over the northern extremities of the Arabian Sea (Figure 4). Due to higher baroclinic instability, winter and spring are favorable seasons for the occurrences of WDs, which are extratropical cyclonic systems that generally originate in the Mediterranean region and are embedded in and moved eastwards by the subtropical westerly jet (SWJ) stream (Dimri and Chevuturi, 2016; Hunt et al., 2018a). These systems are known to bring moisture from the surrounding seas (Bookhagen and Burbank, 2006), mostly from the Mediterranean Sea and the Atlantic Ocean, more accurate estimates of major moisture sources of the ARs, however, can be obtained from Lagrangian trajectory analyses (Nayak et al., 2016; Sodemann and Stohl, 2013). As discussed in the introduction section, the presence of cyclonic systems and jet streams are typical features of ARs. In addition, the poleward component of moisture transport, a typical feature of ARs, is apparent in ARs of all seasons (Figure 4).

WDs and SWJ are mostly absent during summer and autumn and only limited number of ARs travel over the northern Arabian Sea. Most ARs in these seasons travel over the southern Arabian sea and the Bay of Bengal, and their tracks indicate that they may be associated with the Indian Summer Monsoon and systems affecting it, such as the Somali Jet, ITCZ, Tropical Easterly Jet. A detailed association of ARs in summer with the monsoon circulation is a subject for future investigation.

From Figure 4, we also observed ARs that travelled over the northern Arabian Sea are generally of weak categories, and, in contrast, the ARs that travelled over the Bay of Bengal fall generally in strong categories, stronger than Cat 3. This is most likely because ARs over the Bay of Bengal travel over the tropics, where atmosphere is generally hot and humid as compared to northern Arabian sea. Interestingly, it is noticed that the Bay of Bengal ARs mostly have confined region of travel, landfall



exclusively over the southern bins of the transect, and rarely travel to the Indus and northern Ganga

basins. These traits can be relevant in forecasting ARs and their impacts over the Himalayas.

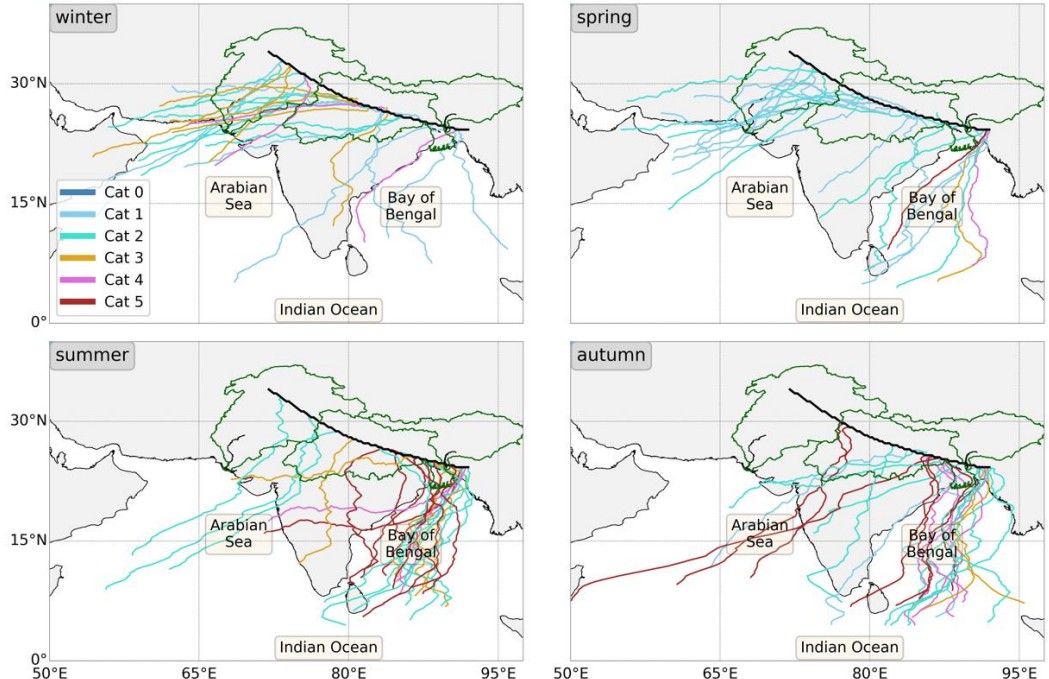

**Figure 4:** Tracks of ARs: Category-wise tracks of ARs during the four seasons. In each season, the tracks of 25 randomly selected ARs are drawn to show the favored paths of ARs. Here, an AR-track is defined as the major axis of the AR, see text for more details.

### 5.3    Precipitation Impacts

The primary goal of this paper is to develop an AR database over the Himalayas; however, it
is important to see whether or not, and to what extent, ARs impact the hydrology of the Himalayan
rivers basins, as they do in other mountainous regions. ARs impact the hydrology of a basin primarily
through the precipitation that is generated when the moisture-laden air is lifted by orography or other
dynamical mechanisms, such as frontal lifting. The purpose of this section is to provide a brief
overview of the impacts ARs can have on the precipitation over the Himalayas, so that their
contribution in shaping the hydrology of the basins may be appreciated in the scientific literature.
Figure 5 shows the observation-based daily precipitation intensities from IMD corresponding to four



long-duration (duration of at least 3 days) and highest intensity ARs, two over each the Indus and the
Ganga basins which span the entire transect of our analysis. In all the four ARs, long and relatively
narrow bands of intense moisture transports are observed over the northern and southern Arabian Sea.
The ARs dumped large quantities of moisture as precipitation along their tracks over the plains and
near the foothills of Himalayas and higher elevations, upon confronting the Himalayan topographic
barrier (represented by the transect in Figure 5). It can be assumed with some certainty that the
topographic barrier vertically lifted the moisture in the ARs, resulting in intense orographic
precipitation. From panels b2 and c2 of Figure 5, we observed that the heavy precipitation events
associated with ARs can occur over higher altitude (4000–5000 m.a.sl.) glacierized regions. ARs, in
general, are warmer than the ambient atmosphere; in this perspective, these results can have profound
impacts on the mass balance and health of glaciers over the Himalayas, which will be discussed further
in the final section.

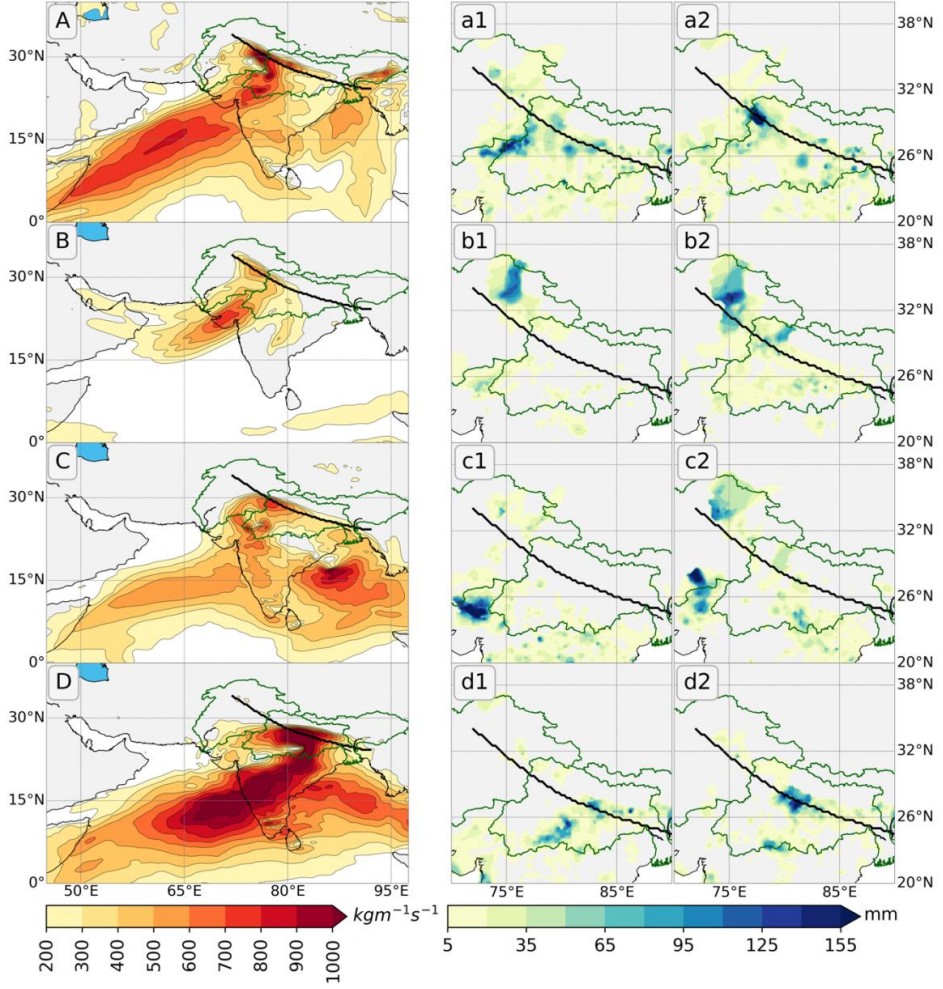

**Figure 5:** Precipitation impacts of ARs: Left panels show IVT fields of the two highest-intensity long-duration
($duration \geq 3 \, days$) ARs over the Indus basin (panel "A" for AR at 1983-07-25 18UTC and panel "B" for
AR at 2003-02-16 06UTC) and over the Ganga basin (panel "C" for AR at 1992-09-06 12UTC, and panel "D"
for AR at 2007-09-21 00UTC). The right panels show the corresponding daily precipitation amounts from IMD
on the second and third day of the AR incidence.
To appreciate the intensity of AR-induced precipitation in a climatological perspective, we
first note that over the Himalayas, the 95th percentile of daily precipitation distribution based on wet
days (defined as days with precipitation larger than 2mm) is roughly 70mm, and the 99th percentile
ranges from 90mm to 125mm. In the right panels of Figure 5, we see that the precipitation intensity
associated with four ARs is larger than 150mm per day, i.e., much higher than the 99th percentile over
a large area, and heavy precipitation crossing the 95th percentile is widespread over thousands of





square kilometers. Such large intensities of precipitation persistently for multiple AR days may
contribute a substantial fraction to the annual accumulated precipitation over the basins. From the
figure, it can also be observed that ARs reach higher altitudes ($> 4000\ m.a.s.l$), where they may
result in severe snowstorms and affect the snow-mass balance of the regions.

We inspected additional six high-intensity ARs in each basin (shown in supplementary

material Figures S2–S3); in majority of them (9 out of 12), widespread heavy precipitation days are
observed. These results indicate that ARs have significant impacts on the water availability, flooding
risks, and glacier health over the Himalayas. Detailed investigations into the hydrological and
ecological impacts of ARs over the Himalayas are left for future studies.
6    Conclusions and Future Prospective

In the present study, for the first time, we constructed a ready-to-use database of ARs over the

Himalayas using ERA5 atmospheric fields since 1982. We used a seasonally varying threshold of IVT
along the Himalayan front in order to extract the ARs. The extracted ARs were further investigated
for their timing, frequency, duration, seasonality, tracks, and trends. The annual frequency of ARs
varied from 15 in 1987 to 35 in 2006 with an average of 23 over 1982–2018 period. The duration of
most ARs (~90%) was less than 3 days, though longer-duration AR are not uncommon. AR occurrence
was most frequent during winter over most of the Himalayan region, except for eastern Ganga and
Brahmaputra basins where highest frequency was observed during summer, which may be linked with
highest summer monsoon activity. We categorized the ARs in six categories, ranging from beneficial
to the most hazardous, based on their intensity and duration. ARs over the Indus Basin, mainly travel
through the Arabian Sea, were mostly of beneficial categories (Cats 0–2), while ARs over the eastern
Ganga and Brahmaputra basins, mainly travel through the Bay of Bengal, were often hazardous
categories (Cats 3–5).

ARs, extreme precipitations, and floods: Recently ARs have been discussed widely for their

societal impacts linked with extreme precipitation in many mountainous regions around the world.



Unfortunately, ARs have been poorly investigated over the Himalayas and consequently the linkages
between ARs and Himalayan hydrology as well as extreme precipitation events are not studied yet.
However, in this study a preliminary investigation between observed ARs and IMD field-based
gridded precipitation data clearly established that ARs are tightly associated with widespread heavy
precipitation days over the Himalayas and can deliver precipitation intensities as high as 150 mm per
day. Though some recent studies effectively linked individual flooding in south India with ARs
(Lakshmi et al., 2019; Laskhmi and Satyanarayana, 2020; Liang and Yong, 2020), but such studies
are scarce over the Himalayan region (Thapa et al., 2018). Our analysis (section 5.3) indicates a strong
association of frequent flooding over the eastern Ganga and Brahmaputra basins with ARs, though
further research is needed to ascertain the association. Recent flood hazards in the Himalayan region
including cloud burst flood in Leh in August 2010 (Dimri et al., 2017), Kedarnath flood in June
2013(Bhambri et al., 2016) and Kashmir flood in September 2014 (Romshoo et al., 2018) have been
discussed as a result of extreme precipitation events but potential linkages of these floods with ARs
have not been explored yet.

ARs and precipitation trends: While the climate warming is unequivocal, long-term precipitation

does not show any particular trend over the past decades in the Himalayan region (Krishnan et al.,
2019). The frequency of ARs over the Himalayas did not show any significant trend in winter and
spring seasons; increasing and decreasing trends, however, are observed in autumn and summer
monsoon seasons, respectively. In northern India, the observed reduced annual precipitations were
thought to be due to weakening of the Indian summer monsoon since 1950s (Mishra et al., 2012; Paul
et al., 2016). We conjecture that the reducing annual precipitations may be associated with decreasing
ARs during the summer monsoon. However, dedicated studies investigating the ARs and monsoon
weakening over the Himalayan region can bring some concrete conclusions on this perspective.

ARs and the Himalayan cryosphere: The Himalayan glaciers are exposed to different climatic

conditions from west to east depending on their geographical location (Maussion et al., 2014).



Numerous field- or model-based studies in the Himalayas advocated that while the combined annual
snow accumulation from the Indian summer monsoon and Western Disturbances shape the glacier
mass balances, it is mainly governed by the snow conditions during the summer monsoon (Azam et
al., 2014a, 2014b; Fujita, 2008; Mandal et al., 2020). Therefore the timing and amount of snowfall
during the melt season are critical factors for the surface energy balance of the Himalayan glaciers,
irrespective of their geographical location. Heavy snowfalls, controlling the glacier mass balances and
meltwater, were simply thought to be rooted in summer-monsoon circulation (Azam et al., 2014b;
Fujita, 2008; Maussion et al., 2014; Mölg et al., 2014). Our study clearly showed ARs' reach to higher
altitudes (up to 4000-5000 m a.s.l.; Figure 5 and Supplementary Figures S2–S3), and we inferr that
due to lower temperatures at higher elevations, ARs have potential to the deliver heavy snowfalls on
the glaciers. Strong snowfalls associated with multiple-day ARs can abruptly change the surface
energy balance on glaciers by increasing surface albedo, especially in the summer melting period, and
thus might be a major driver for glacier mass balance in the Himalayas.

Detailed investigation of the impact of ARs on the Himalayan water resources is beyond the

scope of the present study. Nevertheless, some inferences in our study from the AR dataset developed
offer new lines of research in the Himalayas and invite researchers to investigate the control of ARs
on the health of glaciers, generated melt waters, extreme precipitations, flooding and rain-on-snow
events in the Himalayan basins.

## Data Availability

The data is available at Zenodo repository at https://doi.org/10.5281/zenodo.4451901 (Nayak et al.,
2021). We have also included the data in our Supplementary Information file.

## Acknowledgement

Munir Ahmad Nayak and Rosa Vellosa Lyngwa gratefully acknowledge the financial support provided
by the Science and Engineering Research Board (SERB) of the Department of Science and Technology





(DST), Government of India, under the Early Career Research (ECR) award ECR/2017/002782. We
acknowledge and thank European Centre for Medium-Range Weather Forecasts (ECMWF) for
keeping the data publicly accessible, without which the study would not have been possible.
The authors declare no competing interests.

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
