# Peer review of "ERA5-based database of Atmospheric Rivers over Himalayas"

_Earth System Science Data, 2020_

## Author Comment (AC1)

**Title: 'ERA5-based database of Atmospheric Rivers over Himalayas'**

Author(s): Munir Ahmad Nayak et. al

Manuscript No.: essd-2020-397

Manuscript type: Data description paper

Special Issue: Extreme environment datasets for the three poles

**Referee 3**

The authors created a dataset of atmospheric rivers for the Himalayan region derived from ERA5 data. This dataset could be useful for the community for research into extreme precipitation and flooding. The manuscript is well written and presented.

Response:

We thank the Referee for appreciating our manuscript and his/her thoughtful comments. We agree with the Referee and believe that the dataset will advance AR studies over the unexplored Himalayas.

I think the manuscript could benefit from a bit more explanation on the choice of the AR detection algorithm and the chosen time step. Why did the authors decide to use 6-hourly data despite ERA5 being available on a higher temporal resolution? What made the authors choose this AR identification method over other available methods?

Response:

Thank you for the comments. We have used the modified version of Lavers et al., (2012) algorithm for AR identification in this study as the algorithm is region-specific and allows for a space-time varying threshold criteria. This allows identification of ARs of "weak" and "moderate" ARs in this region in all seasons; for example, in winter season in the western Himalaya, ARs rarely cross the $500 \ kg \ m^{-1}s^{-1}$ (Figure 3) IVT, which will be disqualified by strict high-threshold

algorithms. We wanted to include "weak" ARs in the database as these can have influence on this regional precipitation and important hydrological impacts as observed in some recent studies over other cold regions (Gorodetskaya et al., 2014; Nash et al., 2018; Wille et al., 2019). Also, we would like to highlight that two recent review studies on comparison of AR algorithms (Lora et al., 2020; Rutz et al., 2019) concluded that most of these algorithms identify ARs with fairly good agreement, especially the moderate to intense ARs, and that high-threshold algorithms ignore "weak ARs" from their records. The "weak" ARs are often the instances of ARs formation, dissipation, and merging (Lora et al., 2020), hence can provide important insights into the dynamical evolution of ARs formation and may have important societal impacts.

We have used 6-hourly ERA5 datasets because of four main reasons:

(1) This temporal resolution is commonly used in AR-detection algorithms when using global reanalysis products (Nash & Carvalho, 2020; Waliser & Guan, 2017),

(2) Our main goal is to identify ARs in the Himalayas and provide a ready-to-use and easily-manageable AR database for AR studies over this region for a sufficiently long period i.e., 37 years. We realize that for such a lengthy duration (including all the seasons) to reduce the data volume, 6-hourly analysis is sufficient to produce a distinct and manageable database that can be loaded in most of the software on a home desktop machine. In contrast, 1-hourly AR data will consume more RAM due to larger size, thereby reduce the system performance, while only adding marginal information than 6-hourly data.

(3) 6-hourly datasets provide sufficient temporal information to show the gradual evolution of AR over time (Nash et al., 2018; Ramos et al., 2015), rather than abrupt changes. It is worth mentioning that most climate model simulations for ARs are also archived at this temporal resolution.

(4) A previous study (Rutz et al., 2014) has found similar results in mean AR duration when 6-hourly ERA-Interim IVT dataset is used compared to 1-hourly observational based dataset used by an earlier study (Ralph et al., 2013) for the same study area in Bodega Bay, US West Coast. Another study (Dettinger, 2011) also observed similar results in AR duration when daily observations are used instead of 1-hourly in northern California.

When looking into the dataset I think there could be a bit more additional information on how the data is organised. I am not sure that someone downloading the dataset would be able to understand

it in its current form. For example, it took me a while to figure out that a detected AR has a unique id but still has multiple rows as it consists of multiple timesteps. The description in the read me file is very short and could say more about the structure in the .csv files, e.g. that there is a line for every time step in an identified AR. The manuscript and meta data say that the covered period is 1982-2018 while the first detected AR in the files is from January 1979. For one AR timestep the IVT max says one value but when looking into the columns there is a higher IVT value. It seems a bit complicated organised that the longitudes and latitudes corresponding to the AR locations are in different files from the actual IVT values.

Response

We agree with the Reviewer, perhaps we were not detailed enough. We have updated the readme file, which now reads as:

We have also included a note regarding the 1979 to 1981 ARs, where we mention that cyclone dates have not been removed in these years due to unavailability of cyclone dates, so some cyclones may have been identified as ARs in this period.

-----Readme document text starts here-----

"Atmospheric Rivers (ARs) are long and narrow regions of intense moisture transport in the lower troposphere. The dataset comprises of Atmospheric Rivers that have happened over the Himalayan Basins from 1982 to 2018. It includes the dates and times, duration, intensity/magnitude, tracks, and categories of the ARs.

**File Names and description:**

**1. ERA5_Persistant_Database2000km:** This file includes the date, times, average Integrated Water Vapor Transport (IVT) magnitude ($kg.m^{-1}.s^{-1}$), starting IVT, maximum IVT, and duration of ARs. These terms are explained below in greater details.

**Column "Date":**

Gives the date and time (in Coordinated Universal Time UTC) of each AR timestep. The IVT data used to identify ARs is 6-hourly (00UTC, 06UTC, 12UTC and 18UTC).

**Column "AR_ID":**

Each identified persistent AR, lasting for at least 18 hours, is given a unique ID, which remains same for all timesteps of the AR. This column gives the ID of ARs. The ID of an AR is based on the year in which the AR occurred, the letters "AR", and the occurrence serial of the AR in the year. For example, the first AR in 1990 has ID 1980AR1. If the AR lasted for 10 timesteps, all 10 timesteps will have the same ID.

**Column "Ind":**

This column gives the python index of IVT data in 6-hour yearly data, giving the date and time of each AR timestep. This column can be ignored since the same information is more directly available in "Date" column.

**Column "AvgIVT":**

This column gives the average IVT magnitude $(kg.m^{-1}.s^{-1})$ along the AR major axis, i.e., the gridcells that have maximum IVT along the AR track. For example, the first value corresponds to the average of all values from column *"0"* to column *"88",* which give the IVT magnitude at each gridcell of the major axis of the first timestep.

**Column "StartIVT":**

This column gives the IVT magnitude $(kg.m^{-1}.s^{-1})$ at the initial gridcell on the first timestep when AR condition was identified.

**Column "ARDuration":**

This column gives duration of the AR in hours; for example, an AR lasting for three timesteps will have the duration of 18 hours, an AR lasting for four timesteps will have duration of 24 hours.

**Column "MaxIVT":**

This column gives the maximum of all IVT values $(kg.m^{-1}.s^{-1})$ at the starting gridcells on each timestep of an AR.

**Column "ARCat":**

This column gives category of the AR, based on IVT magnitude and duration of the ARs. Six categories have been defined, Cat0 denoting the weakest AR and Cat5 denoting the strongest AR. More details on this can be found in the accompanying paper.

**Column "0" to the end.**

These columns give the IVT magnitude $(kg.m^{-1}.s^{-1})$ at each gridcell of the major axis of each AR timestep.

*Note that the cyclone dates were not available before 1982, so AR dates for 1979 to 1981 includes cyclonic IVT structures.*

**2. ERA5_Persistant_Database_lats_2000km:** The file gives the latitudes of grid points of maximum IVT, i.e., the latitude of major axes of ARs throughout their duration.

*Columns "Date", "AR_ID", "Ind", "AvgIVT", "StartIVT", "ARDuration", "MaxIVT", "ARCat" are the same as given above for "ERA5_Persistant_Database2000km.csv" file.*

**Column "0" to end.**

These columns give the latitude (in degrees North) at each gridcell of the major axis of each AR timestep.

**3. ERA5_Persistant_Database_lons_2000km:** The file gives the longitudes of grid points of maximum IVT, i.e., the longitudes of major axes of ARs throughout their duration

Columns "Date", "AR_ID", "Ind", "AvgIVT", "StartIVT", "ARDuration", "MaxIVT", "ARCat" are the same as given above for "ERA5_Persistant_Database2000km.csv" file.

**Column "0" to end.**

These columns give the longitude (in degrees East) at each gridcell of the major axis of each AR

timestep"

-----Readme document text end -----

Line 246-247: there is "southward" twice in this sentence, while I think one of them should be "eastward".

Response

      Corrected, thank you for pointing this out.

**References:**

Dettinger, M. D. (2011). Climate change, Atmospheric Rivers, and floods in California - A

    Multimodel analysis of storm frequency and magnitude changes. *JAWRA Journal of the*

    *American Water Resources Association*, *47*(3), 514–523. https://doi.org/10.1111/j.1752-

    1688.2011.00546.x

Gorodetskaya, I. V., Tsukernik, M., Claes, K., Ralph, M. F., Neff, W. D., & Van Lipzig, N. P. M.

    (2014). The role of Atmospheric rivers in anomalous snow accumulation in East

    Antarctica. *Geophysical Research Letters*, *41*(17), 6199–6206.

    https://doi.org/10.1002/2014GL060881

Guan, B., & Waliser, D. E. (2017). Atmospheric rivers in 20 year weather and climate

    simulations: A multimodel, global evaluation. *Journal of Geophysical Research:*

    *Atmospheres*, *122*(11), 5556–5581. https://doi.org/10.1002/2016JD026174

Lavers, D. A., Vallarini, G., Allan, R. P., Wood, E. . F., & Wade, A. J. (2012). The detection of

    atmospheric reanalyses and their links to British winter floods and the large-scale

    climatic circulation. *Journal of Geophysical Research: Atmospheres*, *117*(D20).

    https://doi.org/10.1029/2012JD018027.

Lora, J. M., Shields, C. A., & Rutz, J. J. (2020). Consensus and disagreement in Atmospheric river

    detection: ARTMIP global catalogues. *Geophysical Research Letters*, *47*(20).

    https://doi.org/10.1029/2020GL089302

Nash, D., & Carvalho, L. M. V. (2020). Brief Communication: An electrifying atmospheric river –

    understanding the thunderstorm event in Santa Barbara County during March 2019.

*Natural Hazards and Earth System Sciences*, *20*(7), 1931–1940.

https://doi.org/10.5194/nhess-20-1931-2020

Nash, D., Waliser, D., Guan, B., Ye, H., & Ralph, F. M. (2018). The role of Atmospheric rivers in

extratropical and polar hydroclimate. *Journal of Geophysical Research: Atmospheres*,

*123*(13), 6804–6821. https://doi.org/10.1029/2017JD028130

Ralph, F. M., Coleman, T., Neiman, P. J., Zamora, R. J., & Dettinger, M. D. (2013). Observed

impacts of duration and seasonality of Atmospheric-river landfalls on soil moisture and

runoff in Coastal Northern California. *Journal of Hydrometeorology*, *14*(2), 443–459.

https://doi.org/10.1175/JHM-D-12-076.1

Ralph, F. M., Rutz, J. J., Cordeira, J. M., Dettinger, M. D., Anderson, M., Reynolds, D., et al.

(2019). A scale to characterize the strength and impacts of Atmospheric rivers. *Bulletin

of the American Meteorological Society*, *100*(2), 269–289.

https://doi.org/10.1175/BAMS-D-18-0023.1

Ramos, A. M., Trigo, R. M., Liberato, M. L. R., & Tomé, R. (2015). Daily precipitation extreme

events in the Iberian Peninsula and its sssociation with Atmospheric Rivers. *Journal of

Hydrometeorology*, *16*(2), 579–597. https://doi.org/10.1175/JHM-D-14-0103.1

Rutz, J. J., Steenburgh, W. J., & Ralph, F. M. (2014). Climatological characteristics of

Atmospheric rivers and their inland penetration over the western United States.

*Monthly Weather Review*, *142*(2), 905–921. https://doi.org/10.1175/MWR-D-13-

00168.1

Rutz, J. J., Steenburgh, W. J., & Ralph, F. M. (2015). The inland penetration of Atmospheric

rivers over Western North America: A lagrangian analysis. *Monthly Weather Review*,

*143*(5), 1924–1944. https://doi.org/10.1175/MWR-D-14-00288.1

Rutz, J. J., Shields, C. A., Lora, J. M., Payne, A. E., Guan, B., Ullrich, P., et al. (2019). The

Atmospheric River Tracking Method Intercomparison Project (ARTMIP): Quantifying

uncertainties in Atmospheric river climatology. *Journal of Geophysical Research:*

*Atmospheres*, *124*(24), 13777–13802. https://doi.org/10.1029/2019JD030936

Waliser, D., & Guan, B. (2017). Extreme winds and precipitation during landfall of Atmospheric

rivers. *Nature Geoscience*, *10*(3), 179–183. https://doi.org/10.1038/ngeo2894

Wille, J. D., Favier, V., Dufour, A., Gorodetskaya, I. V., Turner, J., Agosta, C., & Codron, F. (2019).

West Antarctic surface melt triggered by Atmospheric rivers. *Nature Geoscience*, *12*(11),

911–916. https://doi.org/10.1038/s41561-019-0460-1

---

## Author Comment (AC3)

**Title: 'ERA5-based database of Atmospheric Rivers over Himalayas'**

Author(s): Munir Ahmad Nayak et. al

Manuscript No.: essd-2020-397

Manuscript type: Data description paper

Special Issue: Extreme environment datasets for the three poles

*In what follows, we have copied the Reviewers comment verbatim in black color text and have provided our point-by-point response to their comments in blue color.*

**Reviewer 2**

**General Comments**

The manuscript is well written and the concept for this paper is well thought out. The length and structure of the article are appropriate. High Mountain Asia could certainly benefit from more AR analysis due to its unique topography. However, I do not think that this manuscript is ready for publication in its current form. I have several suggestions for improvement of this paper (see below).

Response

The authors would like to sincerely thank the Reviewer for recognizing the importance of our work, for carefully reviewing our work, for the positive feedback, and for the suggestions to improve the quality of our work.

**Specific Comments**

It is unclear how this AR detection algorithm is unique compared to other AR detection algorithms available for Southern Asia. I agree with Reviewer 1 that spatio-temporal availability of ERA5 data could be leveraged for an improved detection algorithm (i.e. 1-hourly, 0.1° horizontal

resolution) in this region. At the very least, the authors could comment on why they chose 6-hourly and 0.25° horizontal resolution.

Response

We agree with the Reviewer that there are indeed many algorithms available to detect ARs. Two recent reviews on comparison of AR algorithms (Lora et al., 2020; Rutz et al., 2019) have highlighted that all AR algorithms provide robust identification of AR features and good agreement in the detection of moderate ($IVT\sim500\ kg\ m^{-1}\ s^{-1}$) and strong ($IVT\sim700\ kg\ m^{-1}\ s^{-1}$) AR events. These reviews, based on in the ARTMIP (Atmospheric River Tracking Method intercomparison Project), also included the algorithm by Lavers et al., (2012), a modified version of which is adopted in the present study. The major differences among these algorithms come from the exclusion or inadequately identifying "weak" AR features ($<$ $250\ kg\ m^{-1}\ s^{-1}$); however, $IVT < 250 kg.\,m^{-1}s^{-1}$ may not be considered "weak" depending on the season and region of interest. The "weak" or less intense ARs are usually found in cold regions like Antarctica, Arctic, and the western Himalaya, or when ARs are just forming or dissipating. These ARs are the ones that account for the major differences among the AR detection algorithms (Lora et al., 2020). Weak ARs are generally excluded by high threshold detection algorithms like Sellars et al., (2017) and Mahoney et al., (2016). For the Himalaya region, we want to identify and include weak ARs in our database as many authors have reported the importance of weak ARs in modulating the hydroclimate of cold regions like Antarctic and Artic (Gorodetskaya et al., 2014; Mattingly et al., 2018; Nash et al., 2018; Wille et al., 2019).

We do not believe that the Lavers et al., (2012) (Lavers) is unique for use over the Himalaya; however, it has a few advantages that appear appealing in the present context. Lavers algorithm is region-specific i.e., a detection transect can be defined precisely at the location required, which will help detect only those ARs that penetrate the Himalayan base. Many algorithms, for example, the Pan & Lu, (2019), require defining a rectangular region and ARs detected within the region may not necessarily impact or cross a specific location of interest. Another advantage of the Lavers algorithm is that it uses climatology-based threshold, dependent on location and season, which can account for the smaller saturation capacity of ARs in cold season over the Himalayas. Many algorithms, such as Sellars et al., (2017) and Liang & Yong, (2020),

use a fixed threshold, $e.g., 750$ or $500\ kg.m^{-1}.s^{-1}$, regardless of season and location, which is too extreme to identify "weak" ARs, most likely present in the Himalayas. Many algorithms (Gershunov et al., 2017; Mahoney et al., 2016; Sellars et al., 2017), which track the life cycle of ARs, are complex, and are not suited for the present work, since our aim is provide a database of ARs, not necessarily their origin, moisture sources, and life cycle. In summary, we preferred the Lavers algorithm mainly for its conceptual and computational simplicity.

**Comment on temporal resolution**

We have used 6-hourly ERA5 datasets because of four main reasons:

(1) This temporal resolution is commonly used in AR-detection algorithms when using global reanalysis products (Nash & Carvalho, 2020; Waliser & Guan, 2017),

(2) Our main goal is to identify ARs in the Himalayas and provide a ready-to-use and easily-manageable AR database for AR studies over this region for a sufficiently long period i.e., 37 years. We realize that for such a lengthy duration (including all the seasons) to reduce the data volume, 6-hourly analysis is sufficient to produce a distinct and manageable database that can be loaded in most of the software on a home desktop machine. In contrast, 1-hourly AR data will consume more RAM due to larger size, thereby reduce the system performance, while only adding marginal information than 6-hourly data. If we carefully assess the benefits of hourly ARs versus 6-hourly ARs, we notice that there are no significant advantages in using hourly observation. For example, since we compute the integrated water vapor transport (IVT) at 4-time steps i.e., 00UTC, 06UTC, 12UTC, and 18 UTC a day, if an AR is identified at any of these time steps say at 06 UTC, five hours before 06 UTC (01, 02, 03, 04, 05 UTC) is considered in the AR duration, but if no AR is identified for one-time step, we will only miss 5-hour analysis in the worst case scenario.

(3) 6-hourly datasets provide sufficient temporal information to show the gradual evolution of AR over time (Nash et al., 2018; Ramos et al., 2015), rather than abrupt changes. It is worth mentioning that most climate model simulations for ARs are also archived at this temporal resolution.

(4) A previous study (Rutz et al., 2014) has found similar results in mean AR duration when 6-hourly ERA-Interim IVT dataset is used compared to 1-hourly observational based dataset used by an earlier study (Ralph et al., 2013) for the same study area in Bodega Bay, US West Coast. Another study (Dettinger, 2011) also observed similar results in AR duration when daily

observations are used instead of 1-hourly in northern California. Other studies (Guan & Waliser, 2015, 2017; Rutz et al., 2014; Shields et al., 2018) have also shown small differences in AR characteristics (frequency, duration, length, etc.) when temporal resolutions of dataset are varied from 6-hourly to 1-hourly or from 6-hourly to 3-hourly or from 6-hourly to 12-hourly.

We have added more information in the revised manuscript (Data Section) to justify the choice of temporal resolution. The text there reads as:

*"The 6-hourly interval is chosen for four main reasons 1.) it is a common denominator among AR detection algorithms using atmospheric reanalysis datasets (Brands et al., 2017; Guan & Waliser, 2015; Mundhenk et al., 2016; Rutz et al., 2014), 2) it provides sufficient temporal information on AR events and captures the gradual changes of AR characteristics (Nash & Carvalho, 2020; Ramos et al., 2015), 3.) many studies have found minor differences in ARs based on differing the temporal resolutions (Guan & Waliser, 2015, 2017; Rutz et al., 2014; Shields et al., 2018), and 4.) as compared to 1-hourly data, it is easily-manageable on a desktop machine with small random access memory (RAM), while marginally compromising on the extent of information available on AR characteristics".*

The authors do note that there are many algorithms available that identify ARs (lines 207-208) but do not employ any comparison with their algorithm and others that are available. For example, other AR detection algorithms on a global, 6-hourly basis (e.g. Guan & Waliser, (2015), (B. Guan et al., (2018), Guan & Waliser, (2019), Sellars et al., (2017) are freely available to the public and could be used for statistical evaluation. This would also give the authors the chance to give error estimates for their data set. Table 1 in Rutz et al (2019) would be a good place to look for available AR detection algorithms in HMA. This would improve the article greatly as it would give the authors a chance to show how novel their algorithm is and why the AR community needs yet another AR detection algorithm. Unless the authors can show that this detection algorithm is better suited for HMA compared to other available AR detection algorithms, this study does not significantly contribute to the current body of work.

Response

Thank you for the detailed suggestions. Though we justify the use of modified version of Lavers et al., 2012 AR-detection algorithm to the study region (in our response to the Reviewer's previous comment and in the revised version of the manuscript), we believe a comparison with a few global algorithms will undoubtedly enhance the quality of the dataset. To this end, we have requested a few authors (Dr. Bin Guan and Dr. Scott Sellers) to share their AR identification codes; we are waiting for their responses. Regardless, since there is an open AR global database based on MERRA, we may still be able to compare ARs detected here with a few global AR-detection algorithms. Accordingly, the comparison results will be added to the revised manuscript.

I would also recommend the author review the ARTMIP articles that complete an in-depth comparison of most of the available AR detection algorithms (Shields et al, 2018; Rutz et al, 2019; Lora et al, 2020) and elaborate on why they chose to emulate the Lavers et al (2012) method over others. For example, why is this method more appropriate for HMA?

Response:

Thank you for the suggestion. We have reviewed a few ARTMIP studies and accordingly added the most relevant references in the revised manuscript, which read as:

*"Many algorithms are available to identify or track ARs. The AR Tracking Method Intercomparison Project (ARTMIP) was initiated to compare different AR algorithms using a common reanalysis dataset (Shields et al., 2018). Rutz et al., (2019) found differences among algorithms when compared in their native configuration (setup), but highlighted the agreements on AR distribution across latitudes in US and Europe when normalized. Lora et al., (2020) then expanded this study globally and found robust agreements among algorithms in identifying "strong" and "moderate" ARs but considerable differences for "weak" ARs. They attributed the disagreement mainly to the high-threshold algorithms that only detect "strong" ARs or identify only the core regions of ARs, while low-threshold algorithms capture overall ARs intensities at different locations, even outside the widely accepted extratropical regions. Here, we modified the algorithm developed by Lavers et al., (2012) to identify ARs over the Himalayas, since the algorithm is conceptually and computationally simple. The algorithm is region specific and allows*

*for the use of space and time varying threshold. The algorithm has been successfully employed in many AR studies over the US West Coast (Barth et al., 2017), Europe (Lavers &Villarini, 2015a, 2015b), and the central US (Lavers &Villarini, 2013a; Nayak et al., 2016; Nayak & Villarini, 2017). For the Himalayas, we want to identify and include "weak" ARs in our database, as they may have important impacts on regional precipitation, as observed by various studies on cold regions like Antarctic and Artic (Gorodetskaya et al., 2014; Mattingly et al., 2018; Nash et al., 2018; Wille et al., 2019)."*

Regarding the choice of Lavers et al., (2012) algorithm for AR identification, we refer the Reviewer to our response to her/his first specific comment.

The author briefly mentions the AR study over the Bay of Bengal (Yang et al, 2018) in the results section, but it is not mentioned in the introduction paragraph where the author discusses other studies that examine ARs in Southern Asia (lines 102-118).

Response

Thank you for noting this. We have added the following points in the revised version of the manuscript:

*"Yang et al., 2018 identified ARs originating in the Bay of Bengal over the period 1979 to 2011 using ERA-Interim reanalysis. It is observed that these ARs made landfall exclusively over the southern foothills of the Himalayas, mainly in Bangladesh, Burma, and occasionally in India. Since the study did not consider ARs originating from the western sources, including the Arabian Sea and the Mediterranean Sea, ARs were not detected in the northern and central Himalaya."*

The readme for the AR track data seems incomplete. I'm not sure the data set would be able to be easily understood and re-used in the future. For example, what do all the columns mean in each of the files? Is there a unique ID for each of the AR tracks, or would a potential user have to join the tables on multiple columns? I would suggest clarification in the readme that describes the columns to prevent misuse of this database in the future.

Response

Thank you for pointing this out; we agree that the readme file may have not been thorough enough. We have now updated it by explaining the data in each column, along with units, and examples wherever necessary. The readme file now reads as:

"Atmospheric Rivers (ARs) are long and narrow regions of intense moisture transport in the lower troposphere. The dataset comprises of ARs that have happened over the Himalayan Basins from 1982 to 2018. It includes the dates and times, duration, intensity/magnitude, tracks, and categories of the ARs.

**File Names and description:**

**1. ERA5_Persistant_Database2000km:** This file includes the date, times, average Integrated Water Vapor Transport (IVT) magnitude ($kg.m^{-1}.s^{-1}$), starting IVT, maximum IVT, and duration of ARs. These terms are explained below in greater details.

**Column "Date":**

Gives the date and time (in Coordinated Universal Time UTC) of each AR timestep. The IVT data used to identify ARs is 6-hourly (00UTC, 06UTC, 12UTC and 18UTC).

**Column "AR_ID":**

Each identified persistent AR, lasting for at least 18 hours, is given a unique ID, which remains same for all timesteps of the AR. This column gives the ID of ARs. The ID of an AR is based on the year in which the AR occurred, the letters "AR", and the occurrence serial of the AR in the year. For example, the first AR in 1990 has ID 1980AR1. If the AR lasted for 10 timesteps, all 10 timesteps will have the same ID.

**Column "Ind":**

This column gives the python index of IVT data in 6-hour yearly data, giving the date and time of each AR timestep. This column can be ignored since the same information is more directly available in "Date" column.

**Column "AvgIVT":**

This column gives the average IVT magnitude ($kg.m^{-1}.s^{-1}$) along the AR major axis, i.e., the gridcells that have maximum IVT along the AR track. For example, the first value corresponds to the average of all values from column *"0"* to column *"88",* which give the IVT magnitude at each gridcell of the major axis of the first timestep.

**Column "StartIVT":**

This column gives the IVT magnitude ($kg.m^{-1}.s^{-1}$) at the initial gridcell on the first timestep when AR condition was identified.

**Column "ARDuration":**

This column gives duration of the AR in hours; for example, an AR lasting for three timesteps will have the duration of 18 hours, an AR lasting for four timesteps will have duration of 24 hours.

**Column "MaxIVT":**

This column gives the maximum of all IVT values ($kg.m^{-1}.s^{-1}$) at the starting gridcells on each timestep of an AR.

**Column "ARCat":**

This column gives category of the AR, based on IVT magnitude and duration of the ARs. Six categories have been defined, Cat0 denoting the weakest AR and Cat5 denoting the strongest AR. More details on this can be found in the accompanying paper.

**Column "0" to the end.**

These columns give the IVT magnitude ($kg.m^{-1}.s^{-1}$) at each gridcell of the major axis of each AR timestep.

*Note that the cyclone dates were not available before 1982, so AR dates for 1979 to 1981 includes cyclonic IVT structures.*

**2. ERA5_Persistant_Database_lats_2000km:** The file gives the latitudes of grid points of maximum IVT, i.e., the latitude of major axes of ARs throughout their duration.

***Columns "Date", "AR_ID", "Ind", "AvgIVT", "StartIVT", "ARDuration", "MaxIVT", "ARCat" are the same as given above for "ERA5_Persistant_Database2000km.csv" file.***

**Column "0" to end.**

These columns give the latitude (in degrees North) at each gridcell of the major axis of each AR timestep.

**3. ERA5_Persistant_Database_lons_2000km:** The file gives the longitudes of grid points of maximum IVT, i.e., the longitudes of major axes of ARs throughout their duration

Columns "Date", "AR_ID", "Ind", "AvgIVT", "StartIVT", "ARDuration", "MaxIVT", "ARCat" are the same as given above for "ERA5_Persistant_Database2000km.csv" file.

**Column "0" to end.**

These columns give the longitude (in degrees East) at each gridcell of the major axis of each AR timestep"

**Technical Corrections**

Line 345-346: The sentence beginning with "The minimum" is confusing to read and should be rewritten for clarity.

Response

Thank you for highlight this. We have rephrased the sentence for better clarity, and it reads as below in the revised manuscript.

*"The minimum number of ARs observed is 15 (in 1987), while the maximum annual frequency is 37( in 2006)."*

Line 287 (and others) The formatting of ð  ð  ð  $^{-1}$ ð  $^{-1}$ $(kgm^{-1}s^{-1})$ is off. For example, there does not appear to be a space between kg and m. This occurs in the supplemental material as well.

Response

Corrected, thank you pointing this out.

The folder containing the Supplemental materials is misspelled as "Supplementary Information".

Response

Corrected, thank you.

The authors would again like to thank the Reviewer for her/his thoughtful suggestions, which have greatly improved the quality of the manuscript.

**References**

Guan B, Waliser D (2015) Detection of atmospheric rivers: Evaluation and application of an algorithm for global studies. Journal of Geophysical Research: Atmospheres 120(24):12,514–12,535

Guan B, Waliser DE (2019) Tracking atmospheric rivers globally: spatial distributions and temporal evolution of life cycle characteristics. Journal of Geophysical Research: Atmospheres 124(23):12,523–12,552

Guan B, Waliser DE, Ralph FM (2018) An intercomparison between reanalysis and dropsonde observations of the total water vapor transport in individual atmospheric rivers. Journal of Hydrometeorology 19(2):321–337

Lavers DA, Villarini G, Allan RP, Wood EF, Wade AJ (2012) The detection of atmospheric rivers in atmospheric reanalyses and their links to british winter floods and the large-scale climatic circulation. Journal of Geophysical Research: Atmospheres 117(D20), DOI 10.1029/2012JD018027

Lora JM, Shields C, Rutz J (2020) Consensus and disagreement in atmospheric river detection: Artmip global catalogues. Geophysical Research Letters 47(20):e2020GL089,302, DOI 10.1029/2020GL089302

Rutz JJ, Shields CA, Lora JM, Payne AE, Guan B, Ullrich P, O'Brien T, Leung LR, Ralph FM, Wehner M, et al (2019) The atmospheric river tracking method intercomparison project (artmip): quantifying uncertainties in atmospheric river climatology. Journal of Geophysical Research: Atmospheres 124(24):13,777– 13,802, DOI 10.1029/2019JD030936

Sellars S, Kawzenuk B, Nguyen P, Ralph F, Sorooshian S (2017) Genesis, pathways, and terminations of intense global water vapor transport in association with large-scale climate patterns. Geophysical Research Letters 44(24):12–465, DOI 10.1002/2017GL075495

Shields CA, Rutz JJ, Leung LY, Ralph FM, Wehner M, Kawzenuk B, Lora JM, McClenny E, Osborne T, Payne AE, et al (2018) Atmospheric river tracking method intercomparison project (artmip): project goals and experimental design. Geoscientific Model Development 11(6):2455–2474

Yang Y, Zhao T, Ni G, Sun T (2018) Atmospheric rivers over the bay of bengal lead to northern indian extreme rainfall. International Journal of Climatology 38(2):1010–1021, DOI 10.1002/joc.5229

**Two corrections to the previous review:**

1) ERA5 has a horizontal resolution of 0.25°, not 0.1°. So, the author should only comment on their choice to use 6-hourly compared to 3-hourly or even hourly temporal resolution.

Response

    Done. This is responded in the first comment under specific comments.

2) The technical correction for Line 287 (and others) should read: The formatting of kg m$^{-1}$ s$^{-1}$ is off in the manuscript. For example, there does not appear to be a space between kg and m. This occurs in the supplemental material as well.

Response

    Done, thank you.

**References:**

Barth, N. A., Villarini, G., Nayak, M. A., & White, K. (2017). Mixed populations and annual flood
frequency estimates in the western United States: The role of atmospheric rivers:
Atmosopheric rivers and west United States floods. *Water Resources Research*, *53*(1),
257–269. https://doi.org/10.1002/2016WR019064

Brands, S., Gutiérrez, J. M., & San-Martín, D. (2017). Twentieth-century atmospheric river
activity along the west coasts of Europe and North America: algorithm formulation,
reanalysis uncertainty and links to atmospheric circulation patterns. *Climate Dynamics*,
*48*(9–10), 2771–2795. https://doi.org/10.1007/s00382-016-3095-6

Dettinger, M. D. (2011). Climate change, Atmospheric Rivers, and floods in California - A
Multimodel analysis of storm frequency and magnitude changes. *JAWRA Journal of the
American Water Resources Association*, *47*(3), 514–523. https://doi.org/10.1111/j.1752-
1688.2011.00546.x

Gershunov, A., Shulgina, T., Ralph, F. M., Lavers, D. A., & Rutz, J. J. (2017). Assessing the climate-
scale variability of atmospheric rivers affecting western North America. *Geophysical
Research Letters*, *44*(15), 7900–7908. https://doi.org/10.1002/2017GL074175

Gorodetskaya, I. V., Tsukernik, M., Claes, K., Ralph, M. F., Neff, W. D., & Van Lipzig, N. P. M.
(2014). The role of Atmospheric rivers in anomalous snow accumulation in East
Antarctica. *Geophysical Research Letters*, *41*(17), 6199–6206.
https://doi.org/10.1002/2014GL060881

Guan, B., & Waliser, D. E. (2015). Detection of Atmospheric rivers: Evaluation and application of

an algorithm for global studies: Detection of Atmospheric rivers. *Journal of Geophysical

Research: Atmospheres*.

Guan, B., & Waliser, D. E. (2017). Atmospheric rivers in 20 year weather and climate

simulations: A multimodel, global evaluation. *Journal of Geophysical Research:

Atmospheres*, *122*(11), 5556–5581. https://doi.org/10.1002/2016JD026174

Guan, B., & Waliser, D. E. (2019). Tracking Atmospheric rivers globally: spatial distributions and

temporal evolution of life cycle characteristics. *Journal of Geophysical Research:

Atmospheres*, *124*(23), 12523–12552. https://doi.org/10.1029/2019JD031205

Guan, B., Waliser, D. E., & Ralph, F. M. (2018). An Intercomparison between reanalysis and

dropsonde observations of the total water vapor transport in individual Atmospheric

rivers. *Journal of Hydrometeorology*, *19*(2), 321–337. https://doi.org/10.1175/JHM-D-

17-0114.1

Guan, Bin, & Waliser, D. E. (2015). Detection of atmospheric rivers: Evaluation and application

of an algorithm for global studies. *Journal of Geophysical Research: Atmospheres*,

*120*(24), 12514–12535. https://doi.org/10.1002/2015JD024257

Lavers, D. A., & Villarini, G. (2013). Atmospheric rivers and flooding over the central United

States. *Journal of Climate*, *26*(20), 7829–7836. https://doi.org/10.1175/JCLI-D-13-

00212.1

Lavers, D. A., & Villarini, G. (2015a). The contribution of Atmospheric rivers to precipitation in

Europe and the United States. *Journal of Hydrology*, *522*, 382–390.

https://doi.org/10.1016/j.jhydrol.2014.12.010

Lavers, D. A., Vallarini, G., Allan, R. P., Wood, E. . F., & Wade, A. J. (2012). The detection of

atmospheric reanalyses and their links to British winter floods and the large-scale

climatic circulation. *Journal of Geophysical Research: Atmospheres*, *117*(D20).

https://doi.org/10.1029/2012JD018027.

Lavers, D. A., Villarini, G., Allan, R. P., Wood, E. F., & Wade, A. J. (2012). The detection of

Atmospheric rivers in atmospheric reanalyses and their links to British winter floods and

the large-scale climatic circulation. *Journal of Geophysical Research: Atmospheres*,

*117*(D20). https://doi.org/10.1029/2012JD018027

Liang, J., & Yong, Y. (2020). Climatology of Atmospheric rivers in the Asian monsoon region.

*International Journal of Climatology*, joc.6729. https://doi.org/10.1002/joc.6729

Lora, J. M., Shields, C. A., & Rutz, J. J. (2020). Consensus and disagreement in Atmospheric river

detection: ARTMIP global catalogues. *Geophysical Research Letters*, *47*(20).

https://doi.org/10.1029/2020GL089302

Mahoney, K., Jackson, D. L., Neiman, P., Hughes, M., Darby, L., Wick, G., et al. (2016).

Understanding the role of Atmospheric rivers in heavy precipitation in the southeast

United States. *Monthly Weather Review*, *144*(4), 1617–1632.

https://doi.org/10.1175/MWR-D-15-0279.1

Mattingly, K. S., Mote, T. L., & Fettweis, X. (2018). Atmospheric river impacts on Greenland Ice

sheet surface mass balance. *Journal of Geophysical Research: Atmospheres*, *123*(16),

8538–8560. https://doi.org/10.1029/2018JD028714

Mundhenk, B. D., Barnes, E. A., & Maloney, E. D. (2016). All-season climatology and variability

of Atmospheric river frequencies over the North Pacific. *Journal of Climate*, *29*(13),

4885–4903. https://doi.org/10.1175/JCLI-D-15-0655.1

Nash, D., & Carvalho, L. M. V. (2020). Brief Communication: An electrifying atmospheric river –

understanding the thunderstorm event in Santa Barbara County during March 2019.

*Natural Hazards and Earth System Sciences*, *20*(7), 1931–1940.

https://doi.org/10.5194/nhess-20-1931-2020

Nash, D., Waliser, D., Guan, B., Ye, H., & Ralph, F. M. (2018). The role of Atmospheric rivers in

extratropical and polar hydroclimate. *Journal of Geophysical Research: Atmospheres*,

*123*(13), 6804–6821. https://doi.org/10.1029/2017JD028130

Nayak, M. A., & Villarini, G. (2017). A long-term perspective of the hydroclimatological impacts

of atmospheric rivers over the central United States. *Water Resources Research*, *53*(2),

1144–1166.

Nayak, M. A., Villarini, G., & Bradley, A. A. (2016). Atmospheric rivers and rainfall during NASA's

Iowa Flood Studies (IFloodS) Campaign. *Journal of Hydrometeorology*, *17*(1), 257–271.

https://doi.org/10.1175/JHM-D-14-0185.1

Pan, M., & Lu, M. (2019). A Novel Atmospheric River Identification Algorithm. *Water Resources

Research*, *55*(7), 6069–6087. https://doi.org/10.1029/2018WR024407

Ralph, F. M., Coleman, T., Neiman, P. J., Zamora, R. J., & Dettinger, M. D. (2013). Observed

impacts of duration and seasonality of Atmospheric-river landfalls on soil moisture and

runoff in Coastal Northern California. *Journal of Hydrometeorology*, *14*(2), 443–459.

https://doi.org/10.1175/JHM-D-12-076.1

Ramos, A. M., Trigo, R. M., Liberato, M. L. R., & Tomé, R. (2015). Daily precipitation extreme

    events in the Iberian Peninsula and its sssociation with Atmospheric Rivers. *Journal of*

    *Hydrometeorology*, *16*(2), 579–597. https://doi.org/10.1175/JHM-D-14-0103.1

Rutz, J. J., Steenburgh, W. J., & Ralph, F. M. (2014). Climatological characteristics of

    Atmospheric rivers and their inland penetration over the western United States.

    *Monthly Weather Review*, *142*(2), 905–921. https://doi.org/10.1175/MWR-D-13-

    00168.1

Rutz, J. J., Shields, C. A., Lora, J. M., Payne, A. E., Guan, B., Ullrich, P., et al. (2019). The

    Atmospheric River Tracking Method Intercomparison Project (ARTMIP): Quantifying

    uncertainties in Atmospheric river climatology. *Journal of Geophysical Research:*

    *Atmospheres*, *124*(24), 13777–13802. https://doi.org/10.1029/2019JD030936

Sellars, S. L., Kawzenuk, B., Nguyen, P., Ralph, F. M., & Sorooshian, S. (2017). Genesis, pathways,

    and terminations of intense global water vapor transport in association with large-scale

    climate patterns. *Geophysical Research Letters*, *44*(24).

    https://doi.org/10.1002/2017GL075495

Shields, C. A., & Kiehl, J. T. (2016). Simulating the Pineapple Express in the half degree

    Community Climate System Model, CCSM4. *Geophysical Research Letters*, *43*(14), 7767–

    7773. https://doi.org/10.1002/2016GL069476

Shields, C. A., Rutz, J. J., Leung, L. Y., Ralph, F. M., Wehner, M., Kawzenuk, B., et al. (2018).

    Atmospheric River Tracking Method Intercomparison Project (ARTMIP): project goals

    and experimental design. *Geoscientific Model Development*, *11*(6), 2455–2474.

    https://doi.org/10.5194/gmd-11-2455-2018

Waliser, D., & Guan, B. (2017). Extreme winds and precipitation during landfall of Atmospheric

rivers. *Nature Geoscience*, *10*(3), 179–183. https://doi.org/10.1038/ngeo2894

Wille, J. D., Favier, V., Dufour, A., Gorodetskaya, I. V., Turner, J., Agosta, C., & Codron, F. (2019).

West Antarctic surface melt triggered by Atmospheric rivers. *Nature Geoscience*, *12*(11),

911–916. https://doi.org/10.1038/s41561-019-0460-1

---

## Author Comment (AC5)

**Title: 'ERA5-based database of Atmospheric Rivers over Himalayas'**

Author(s): Munir Ahmad Nayak et. al

Manuscript No.: essd-2020-397

Manuscript type: Data description paper

Special Issue: Extreme environment datasets for the three poles

Note: The Reviewer comments are copied below verbatim in black color, along with our point-by-point responses in blue color.

**Reviewer 1**

Dear Reviewer:

Thank you for giving your time to review our work and for your insightful comments and suggestions that have significantly improved the quality of our manuscript.

**General comments:**

The manuscript is well written and provides an interesting topic of detecting atmospheric rivers (ARs) over Himalayas. However, I think the manuscript is more suitable for a climate research journal (e.g., Journal of climate, International journal of climatology) rather than a data journal. Because Atmospheric river is a character of water vapor transport in the atmosphere, like an index, can be calculated from different data (reanalysis, simulations) and difficult to evaluate the accuracy. And data quality is one of the key standard of the current journal.

Response:

We thank the Reviewer for his/her comment. Although vertically integrated vapor transport (IVT) is calculated using reanalysis data, we do not believe ARs are "calculated", rather they are "detected" i.e., "identified" or/and "tracked". This difference is important in differentiating an

index from a dataset. Unlike an index, (e.g., Standardized Precipitation Index or Normalized Vegetation Index, which has one fixed value for a particular day or month, ARs at each timestep form a dataset; at each timestep it includes many characteristics, e. g., its major axis, location, the intensity at each location, its track, its structure, etc. Therefore, we respectfully disagree in labelling an AR as an "index", and strongly believe that our developed dataset of ARs is well suited for ESSD. In fact, since the inception of ARs dataset development, the manuscript has been designed specifically for ESSD.

While replying to Reviewer's other comments below, we provide more details for understanding and recognizing the challenges in AR dataset.

This work calculates the ARs only based on ERA5, i would like to know how it differs if calculated based on other reanalysis data, e.g. MERRA, NCEP, JRA. Do they got the similar results? Which is the best? Such questions need to be answered if the data aims to practical use.

Response:

Several studies have noted that AR detection based on different reanalysis products generally produces similar results in identifying ARs (such as the frequency, length, duration, etc.). We highlight the relevant conclusions from three recent works below.

Nayak & Villarini, (2017), after comparing six reanalysis products (20CRV2, NCEP-NCAR, JRA-55, MERRA, ERA-Interim and NCEP-DOE), concluded that all the reanalysis products generally provided robust identification of AR frequency, along with their characteristics. Similar results were obtained by Brands et al., (2017), who compared ARs from NOAA-CIRES Twentieth Century Reanalysis v2 (NOAA-20C) and ECMWF ERA-20C (ERA-20C) over multiple regions of the globe. They noted that though the difference (using bias and correlation) between the frequency of ARs based on the two products is significant in the early 20th century, it approaches almost zero towards the end of the century, and the correlations between the frequencies are near 1 (see their Figures 7 and 8). Over multiple regions of the globe, Bin Guan & Waliser, (2015) noted that ERA-Interim reanalysis-based and MERRA reanalysis-based ARs showed similar characteristics, with more than 91% temporal matching in their occurrences (see their Figure 5J and L).

Having said that, we believe and agree with the reviewer that studies on comparing ARs based on different reanalysis and detection algorithms can be valuable for forwarding the AR science. However, such inter-comparison analysis needs dedicated studies. As a matter of fact, this is the main goal of the Atmospheric River Tracking Method Project (ARTMIP).

Following the Reviewer's concern, we have added the below text in revising the manuscript on intercomparison of AR data based on different reanalysis and different identification algorithms.

*"It may of interest to compare the present ERA5-based AR dataset with ARs from other different reanalysis products and identified using different algorithms, a major objective the Atmospheric River Tracking Method Project (ARTMIP, https://www.cgd.ucar.edu/projects/artmip/). However, since many recent studies have concluded a robust identification of ARs (Nayak & Villarini, (2017), Brands et al., (2017), Guan & Waliser, (2015), Lora et al., 2020, Rutz et al, 2019 ), we believe the comparison may not help in improving the quality of the dataset and may deviate the reader from the main objective of the manuscript."*

Further, ERA5 has hourly resolution, why you use six hourly data? This has disadvantages: 1, it will reduce the accuracy of detecting durations of ARs. 2, if you use instantaneous wind speed to calculate u*q, four times a day could largely differs from 24 times a day, because the atmospheric vapor transport has strong diurnal cycle, especially during monsoon season.

Response:

We thank the Referee for her/his comment. We have used 6-hourly ERA5 datasets because of four main reasons:

(1) This temporal resolution is commonly used in AR-detection algorithms when using global reanalysis products (Nash & Carvalho, 2020; Waliser & Guan, 2017). It is worth mentioning that most climate model simulations for ARs are also archived at this temporal resolution.

(2) Our main goal is to identify ARs in the Himalayas and provide a ready-to-use and easily manageable AR database for AR studies over this region for a sufficiently long period i.e., 37 years. We realize that for such a lengthy duration (including all the seasons) to reduce the data

volume, 6-hourly analysis is sufficient to produce a distinct and manageable database that can be loaded in most of the software on a home desktop machine. In contrast, 1-hourly AR data will consume more RAM due to larger size, thereby reduce the system performance, while only adding marginal information than 6-hourly data. If we carefully assess the benefits of hourly ARs versus 6-hourly ARs, we notice that there are no significant advantages in using hourly observation. For example, since we compute the integrated water vapor transport (IVT) at 4-time steps i.e., 00UTC, 06UTC, 12UTC, and 18 UTC a day, if an AR is identified at any of these time steps say at 06 UTC, five hours before 06 UTC (01, 02, 03, 04, 05 UTC) is considered in the AR duration, but if no AR is identified for one-time step, we will only miss 5-hour analysis in the worst-case scenario.

(3) 6-hourly datasets provide sufficient temporal information to show the gradual evolution of AR over time (Nash et al., 2018; Ramos et al., 2015), rather than abrupt changes.

(4) A previous study (Rutz et al., 2014) has found similar results in mean AR duration when 6-hourly ERA-Interim IVT dataset is used compared to 1-hourly observational-based dataset used by an earlier study (Ralph et al., 2013) for the same study area in Bodega Bay, US West Coast. Another study (Dettinger, 2011) also observed similar results in AR duration when daily observations are used instead of 1-hourly in northern California. Other studies (Guan & Waliser, 2017; Bin Guan & Waliser, 2015; Rutz et al., 2014; Shields et al., 2018) have also shown small differences in AR characteristics (frequency, duration, length, etc.) when temporal resolution of datasets are varied from 6-hourly to 1-hourly or from 6-hourly to 3-hourly or from 6-hourly to 12-hourly.

We have added more information in the revised manuscript (Data Section) to justify the choice of temporal resolution. The text there reads as:

*"The 6-hourly interval is chosen for four main reasons 1.) it is a common denominator among AR detection algorithms using atmospheric reanalysis datasets (Brands et al., 2017; Bin Guan & Waliser, 2015; Mundhenk et al., 2016; Rutz et al., 2014), 2) it provides sufficient temporal information on AR events and captures the gradual changes of AR characteristics (Nash & Carvalho, 2020; Ramos et al., 2015), 3.) many studies have found minor differences in ARs based on differing the temporal resolutions (Guan & Waliser, 2017; Guan & Waliser, 2015; Rutz et al., 2014; Shields et al., 2018), and 4.) as compared to 1-hourly data, it is easily-manageable on a*

*desktop machine with small random access memory (RAM), while marginally compromising on the extent of information available on AR characteristics".*

Yes, we agree that atmospheric vapor transport, along with local temperature, exhibits diurnal cycles (Fletcher et al., 2020). With ARs, however, this cycle is generally only marginal, because the source of moisture is most often non-local and related to synoptic-scale features, such as extratropical cyclones and low-level jets. In Ramos et al., (2015), for example, we notice that during all the timesteps of an AR, IVT intensity remained above $300\ kg.m^{-1}.s^{-1}$ and varied consistently between $800 - 1000\ kg.m^{-1}.s^{-1}$ in the core of the AR (See Figure 6 in Ramos et al., (2015)). Similar observations can be made in other studies (see, for example, Nash & Carvalho, 2020).

The atmospheric vapor transport in ARs is captured reasonably well using 6-hourly intervals, as these time intervals produce maps that show the gradual transformation of AR intensity over time (changes in AR intensity i.e., broadening or weakening or dissipation) (Neiman et al., 2008).

There are many methods to calculate ARs, which will lead to multiple different results when calculating ARs. The method in your manuscripts itself also has some empirical treatments. For example, line 222, why you use 15 days moving average instead of 10 or 20? Is it physical mechanism dependent? The methods and the based data source selected could lead to large uncertainties in the ARs data.

Response:

We agree with the Reviewer that there are indeed many algorithms available to detect ARs. Two recent reviews on comparison of AR algorithms (Lora et al., 2020; Rutz et al., 2019) have highlighted that all AR algorithms provide robust identification of AR features and good agreement in the detection of moderate $(IVT{\sim}500\ kg\ m^{-1}\ s^{-1})$ and strong $(IVT{\sim}700\ kg\ m^{-1}\ s^{-1})$ AR events. These reviews, based on the ARTMIP (Atmospheric River Tracking Method intercomparison Project), also included the algorithm by (Lavers et al., 2012), a modified version of which is adopted in the present study. The major differences among these

algorithms come from the exclusion or inadequately identifying "weak" AR features ($<$ $250\ kg\ m^{-1}\ s^{-1}$); however, $IVT < 250 kg.m^{-1}s^{-1}$ may not be considered "weak" depending on the season and region of interest. The "weak" or less intense ARs are usually found in cold regions like Antarctica, Arctic, and the western Himalaya, or when ARs are just forming or dissipating. These ARs are the ones that account for the major differences among the AR detection algorithms (Lora et al., 2020). Weak ARs are generally excluded by high threshold detection algorithms like Sellars et al., (2017) and Mahoney et al., (2016). For the Himalayan region, we want to identify and include weak ARs in our database as many authors have reported the importance of weak ARs in modulating the hydroclimate of cold regions like Antarctic and Artic (Gorodetskaya et al., 2014; Mattingly et al., 2018; Nash et al., 2018; Wille et al., 2019).

We do not believe that the Lavers et al., (2012) (Lavers) is unique for use over the Himalayas; however, it has a few advantages that appear appealing in the present context. Lavers algorithm is region-specific i.e., a detection transect can be defined precisely at the location required, which will help detect only those ARs that penetrate the Himalayan base. Many algorithms, for example, the Pan & Lu, (2019), require defining a rectangular region and ARs detected within the region may not necessarily impact or cross a specific location of interest. Another advantage of the Lavers algorithm is that it uses a climatology-based threshold, dependent on location and season, which can account for the smaller saturation capacity of ARs in cold season over the Himalayas. Many algorithms, such as Sellars et al., (2017) and Liang & Yong, (2020), use a fixed threshold, $e.g.,$ 750 or 500 $kg.m^{-1}.s^{-1}$, regardless of season and location, which is too extreme to identify "weak" ARs, most likely present in the Himalayas. Many algorithms (Gershunov et al., 2017; Mahoney et al., 2016; Sellars et al., 2017), which track the life cycle of ARs, are complex, and are not suited for the present work, since our aim is provide a database of ARs, not necessarily their origin, moisture sources, and life cycle. In summary, we preferred the Lavers algorithm mainly for its conceptual and computational simplicity.

We have added more information in the revised manuscript. The text there reads as:

*"Many algorithms are available to identify or track ARs. The AR Tracking Method Intercomparison Project (ARTMIP) was initiated to compare different AR algorithms using a common reanalysis dataset (Shields et al., 2018). Rutz et al., (2019) found differences among*

*algorithms when compared in their native configuration (setup), but highlighted the agreements on AR distribution across latitudes in US and Europe when normalized. Lora et al., (2020) then expanded this study globally and found robust agreements among algorithms in identifying "strong" and "moderate" ARs but considerable differences for "weak" ARs. They attributed the disagreement mainly to the high-threshold algorithms that only detect "strong" ARs or identify only the core regions of ARs, while low-threshold algorithms capture overall ARs intensities at different locations, even outside the widely accepted extratropical regions. Here, we modified the algorithm developed by Lavers et al., (2012) to identify ARs over the Himalayas, since the algorithm is conceptually and computationally simple. The algorithm is region-specific and allows for the use of space and time varying thresholds. The algorithm has been successfully employed in many AR studies over the US West Coast (Barth et al., 2017), Europe (Lavers & Villarini, 2015a, 2015b), and the central US (Lavers & Villarini, 2013; Nayak et al., 2016; Nayak & Villarini, 2017). For the Himalayas, we want to identify and include "weak" ARs in our database, as they may have important impacts on regional precipitation, as observed by various studies in cold regions like Antarctic and Artic (Gorodetskaya et al., 2014; Mattingly et al., 2018; Nash et al., 2018; Wille et al., 2019)."*

We would like to highlight that all AR-identification algorithms have empirical parameters, see the table below adapted from Rutz et al., (2019) and Shields et al., (2018).

| AR-Identification Algorithm name | Parameters | References |
|---|---|---|
| AR Detection Methodology (ARTD) Goldenson | Absolute IVT threshold: $250\ kg\ m^{-1}s^{-1}$
 Absolute IWV threshold > $15\ mm$
 Length $\geq 1500\ km$
 18 hours (3-time steps for 6-hourly) | (Gershunov et al., 2017) |
| Guan_Walis | Relative IVT threshold: latitude dependent 85th percentile threshold.
 Absolute minimum IVT threshold: $100\ kg\ m^{-1}s^{-1}$ for polar locations
 Length: >2000 km,
 Length/Width ratio: >2 | (Bin Guan & Waliser, 2015) |

| | | |
|---|---|---|
| | IVT direction within 45° of AR shape orientation, poleward direction | |
| Lavers | Relative IVT threshold: latitude dependent 85th percentile threshold. 4.5° latitude movement allowed | (Lavers et al., 2012) |
| Ramos | Relative IVT threshold: Latitude dependent 85th percentile Length $\geq 1500\ km$ < 4.5° latitude movement allowed Persistent ARs: 18 hours | (Ramos et al., 2016) |
| Payne & Magnusdittir | Relative IVT threshold: 85th percentile of maximum IVT Absolute IWV threshold: >2cm Persistent AR: $\geq 12\ hour$ 850 hPa, zonal and meridional winds should be positive and $> 10\ m\ s^{-1}$ Length: $2000\ km$ (the central AR axis) in the zonal direction for landfalling only | (Payne & Magnusdottir, 2015) |
| Pan & Lu | Dual threshold (relative): Regional: 80% quantile IVT Local threshold: spatially smooth 85% quantile IVT using Gaussian Kernel density technique Length: $> 2000\ km$ Length/Width ratio: $> 2$ | (Pan & Lu, 2019) |
| Mundhenk | Relative: Latitude dependent IVT percentiles Length: $> 1400\ km$ Aspect ratio: 1:4 Latitudinal limit: 16 N/S | (Mundhenk et al., 2016) |

Now, regarding the 15-day-based IVT threshold, we would like to note that the 15-day moving average is considered only to obtain the threshold for a particular day. This 15-day moving average, based on the previous study by Nayak & Villarini, (2017), is used to smoothen the IVT data against any short-term fluctuations, i.e., to exclude the effect of any outlier and/or noise before computing the daily- varying threshold (smooth threshold). We noted that the choice of 10 or 20-day instead of the 15-day moving average threshold does not significantly impact the threshold as

shown in Figure R1. Many other studies have used moving averages to remove small-scale perturbations in the data before AR analysis (Dettinger, 2016; Komatsu et al., 2018; Matthews et al., 2018; Payne & Magnusdottir, 2014; Xu et al., 2020).

[Figure]

Figure R1: The daily 85th percentile threshold for each bin is shown by different colors (legend on top right) using different moving average constant i.e., 10, 15 and 20-day moving average which is shown by different linestyle (legend on top left).

Why is the threshold of IVT set to different values for different seasons? This will has disadvantages for practical application during disaster research. In such circumstance, for example, you will likely exclude some ARs that will leads to flood in wet season but include some ARs that may not leads to flood in dry season.

Response:

     We understand the Reviewer's concern. We have used a time-varying threshold as it provides a meaningful estimate of a points of change (seasonality) in the natural system, with respect to the local climatology, which can have socio-economic impacts and in general influence the regional weather. Since ARs are anomalous moisture transports beyond the regional

climatology, the use of time-varying and location-dependent threshold is strongly advocated in the AR literature (Brands et al., 2017; Guan & Waliser, 2015; Lavers et al., 2012; Lavers & Villarini, 2013; Mundhenk et al., 2016), mainly due to large inter-seasonal variability in IVT (Figure R1). By using a large, fixed threshold, adopted in some "strict" algorithms such as Sellars et al., (2017)., we may not find ARs in any season over the Himalayas, except in summer. As in many previous studies over other regions (see for Example Lavers & Villarini, (2013)), we reveal the existence of ARs in winter over the Himalayas, and we note their potential in causing extreme precipitation events. We believe a time-varying threshold is reasonable and required for the Himalayas, since it allows the detection of "weak" to "moderate" ARs (Lora et al., 2020) that can be of importance in the complex Himalayan terrain.

**typing error**

Line 31:Driver-like => should it be 'river-like'

Response

Yes, it is "river-like", thank you for pointing it out. We have corrected this in the revised submission.

**References:**

Barth, N. A., Villarini, G., Nayak, M. A., & White, K. (2017). Mixed populations and annual flood frequency estimates in the western United States: The role of Atmospheric rivers. *Water Resources Research*, *53*(1), 257–269. https://doi.org/10.1002/2016WR019064

Brands, S., Gutiérrez, J. M., & San-Martín, D. (2017). Twentieth-century Atmospheric river activity along the west coasts of Europe and North America: algorithm formulation, reanalysis uncertainty and links to atmospheric circulation patterns. *Climate Dynamics*, *48*(9–10), 2771–2795. https://doi.org/10.1007/s00382-016-3095-6

Dettinger, M. (2016). Historical and future relations between large storms and droughts in California. *San Francisco Estuary and Watershed Science*, *14*(2). https://doi.org/10.15447/sfews.2016v14iss2art1

Dettinger, M. D. (2011). Climate change, Atmospheric rivers, and floods in California - A multimodel analysis of storm frequency and magnitude changes. *JAWRA Journal of the American Water Resources Association*, *47*(3), 514–523. https://doi.org/10.1111/j.1752-1688.2011.00546.x

Fletcher, J. K., Parker, D. J., Turner, A. G., Menon, A., Martin, G. M., Birch, C. E., et al. (2020). The dynamic and thermodynamic structure of the monsoon over southern India: New observations from the INCOMPASS IOP. *Quarterly Journal of the Royal Meteorological Society*, *146*(731), 2867–2890. https://doi.org/10.1002/qj.3439

Gershunov, A., Shulgina, T., Ralph, F. M., Lavers, D. A., & Rutz, J. J. (2017). Assessing the climate-scale variability of Atmospheric rivers affecting western North America. *Geophysical Research Letters*, *44*(15), 7900–7908. https://doi.org/10.1002/2017GL074175

Gorodetskaya, I. V., Tsukernik, M., Claes, K., Ralph, M. F., Neff, W. D., & Van Lipzig, N. P. M. (2014). The role of Atmospheric rivers in anomalous snow accumulation in East Antarctica. *Geophysical Research Letters*, *41*(17), 6199–6206. https://doi.org/10.1002/2014GL060881

Guan, B., & Waliser, D. E. (2017). Atmospheric rivers in 20 year weather and climate simulations: A multimodel, global evaluation. *Journal of Geophysical Research: Atmospheres*, *122*(11), 5556–5581. https://doi.org/10.1002/2016JD026174

Guan, B., & Waliser, D. E. (2015). Detection of atmospheric rivers: Evaluation and application of an algorithm for global studies. *Journal of Geophysical Research: Atmospheres*, *120*(24), 12514–12535. https://doi.org/10.1002/2015JD024257

Komatsu, K. K., Alexeev, V. A., Repina, I. A., & Tachibana, Y. (2018). Poleward upgliding Siberian Atmospheric rivers over sea ice heat up Arctic upper air. *Scientific Reports*, *8*(1), 2872. https://doi.org/10.1038/s41598-018-21159-6

Lavers, D. A., & Villarini, G. (2013). Atmospheric rivers and flooding over the central United States. *Journal of Climate*, *26*(20), 7829–7836. https://doi.org/10.1175/JCLI-D-13-00212.1

Lavers, D. A., & Villarini, G. (2015a). The contribution of Atmospheric rivers to precipitation in Europe and the United States. *Journal of Hydrology*, *522*, 382–390. https://doi.org/10.1016/j.jhydrol.2014.12.010

Lavers, D. A., & Villarini, G. (2015b). The relationship between daily European precipitation and measures of Atmospheric water vapour transport. *International Journal of Climatology*, *35*(8), 2187–2192. https://doi.org/10.1002/joc.4119

Lavers, D. A., Villarini, G., Allan, R. P., Wood, E. F., & Wade, A. J. (2012). The detection of

Atmospheric rivers in atmospheric reanalyses and their links to British winter floods and

the large-scale climatic circulation. *Journal of Geophysical Research: Atmospheres*,

*117*(D20). https://doi.org/10.1029/2012JD018027

Liang, J., & Yong, Y. (2020). Climatology of Atmospheric rivers in the Asian monsoon region.

*International Journal of Climatology*, *41*(S1), E801–E818.

https://doi.org/10.1002/joc.6729

Lora, J. M., Shields, C. A., & Rutz, J. J. (2020). Consensus and disagreement in Atmospheric river

detection: ARTMIP global catalogues. *Geophysical Research Letters*, *47*(20).

https://doi.org/10.1029/2020GL089302

Mahoney, K., Jackson, D. L., Neiman, P., Hughes, M., Darby, L., Wick, G., et al. (2016).

Understanding the role of Atmospheric rivers in heavy precipitation in the southeast

United States. *Monthly Weather Review*, *144*(4), 1617–1632.

https://doi.org/10.1175/MWR-D-15-0279.1

Matthews, T., Murphy, C., McCarthy, G., Broderick, C., & Wilby, R. L. (2018). Super Storm

Desmond: a process-based assessment. *Environmental Research Letters*, *13*(1), 014024.

https://doi.org/10.1088/1748-9326/aa98c8

Mattingly, K. S., Mote, T. L., & Fettweis, X. (2018). Atmospheric river impacts on Greenland Ice

sheet surface mass balance. *Journal of Geophysical Research: Atmospheres*, *123*(16),

8538–8560. https://doi.org/10.1029/2018JD028714

Mundhenk, B. D., Barnes, E. A., & Maloney, E. D. (2016). All-season climatology and variability of Atmospheric river frequencies over the North Pacific. *Journal of Climate*, *29*(13), 4885–4903. https://doi.org/10.1175/JCLI-D-15-0655.1

Nash, D., & Carvalho, L. M. V. (2020). Brief Communication: An electrifying Atmospheric river – understanding the thunderstorm event in Santa Barbara County during March 2019. *Natural Hazards and Earth System Sciences*, *20*(7), 1931–1940. https://doi.org/10.5194/nhess-20-1931-2020

Nash, D., Waliser, D., Guan, B., Ye, H., & Ralph, F. M. (2018). The role of Atmospheric rivers in extratropical and polar hydroclimate. *Journal of Geophysical Research: Atmospheres*, *123*(13), 6804–6821. https://doi.org/10.1029/2017JD028130

Nayak, M. A., & Villarini, G. (2017). A long-term perspective of the hydroclimatological impacts of Atmospheric rivers over the central United States. *Water Resources Research*, *53*(2), 1144–1166. https://doi.org/10.1002/2016WR019033

Nayak, M. A., Villarini, G., & Bradley, A. A. (2016). Atmospheric rivers and rainfall during NASA's Iowa Flood Studies (IFloodS) Campaign. *Journal of Hydrometeorology*, *17*(1), 257–271. https://doi.org/10.1175/JHM-D-14-0185.1

Neiman, P. J., Ralph, F. M., Wick, G. A., Lundquist, J. D., & Dettinger, M. D. (2008). Meteorological characteristics and overland precipitation impacts of Atmospheric rivers affecting the West Coast of North America based on eight years of SSM/I Satellite observations. *Journal of Hydrometeorology*, *9*(1), 22–47. https://doi.org/10.1175/2007JHM855.1

Pan, M., & Lu, M. (2019). A Novel Atmospheric river identification algorithm. *Water Resources Research*, *55*(7), 6069–6087. https://doi.org/10.1029/2018WR024407

Payne, A. E., & Magnusdottir, G. (2014). Dynamics of landfalling Atmospheric rivers over the North Pacific in 30 Years of MERRA Reanalysis. *Journal of Climate*, *27*(18), 7133–7150. https://doi.org/10.1175/JCLI-D-14-00034.1

Payne, A. E., & Magnusdottir, G. (2015). An evaluation of Atmospheric rivers over the North Pacific in CMIP5 and their response to warming under RCP 8.5. *Journal of Geophysical Research: Atmospheres*, *120*(21), 11–173. https://doi.org/10.1002/2015JD023586

Ralph, F. M., Coleman, T., Neiman, P. J., Zamora, R. J., & Dettinger, M. D. (2013). Observed impacts of duration and seasonality of Atmospheric-river landfalls on soil moisture and runoff in Coastal Northern California. *Journal of Hydrometeorology*, *14*(2), 443–459. https://doi.org/10.1175/JHM-D-12-076.1

Ramos, A. M., Trigo, R. M., Liberato, M. L. R., & Tomé, R. (2015). Daily precipitation extreme events in the Iberian Peninsula and its sssociation with Atmospheric rivers. *Journal of Hydrometeorology*, *16*(2), 579–597. https://doi.org/10.1175/JHM-D-14-0103.1

Ramos, A. M., Nieto, R., Tomé, R., Gimeno, L., Trigo, R. M., Liberato, M. L. R., & Lavers, D. A. (2016). Atmospheric rivers moisture sources from a Lagrangian perspective. *Earth System Dynamics*, *7*(2), 371–384. https://doi.org/10.5194/esd-7-371-2016

Rutz, J. J., Steenburgh, W. J., & Ralph, F. M. (2014). Climatological characteristics of Atmospheric rivers and their inland penetration over the western United States. *Monthly Weather Review*, *142*(2), 905–921. https://doi.org/10.1175/MWR-D-13-00168.1

Rutz, J. J., Shields, C. A., Lora, J. M., Payne, A. E., Guan, B., Ullrich, P., et al. (2019). The

Atmospheric River Tracking Method Intercomparison Project (ARTMIP): Quantifying

uncertainties in Atmospheric river climatology. *Journal of Geophysical Research:*

*Atmospheres*, *124*(24), 13777–13802. https://doi.org/10.1029/2019JD030936

Sellars, S. L., Kawzenuk, B., Nguyen, P., Ralph, F. M., & Sorooshian, S. (2017). Genesis, pathways,

and terminations of intense global water vapor transport in association with large-scale

climate patterns. *Geophysical Research Letters*, *44*(24), 12–465.

https://doi.org/10.1002/2017GL075495

Shields, C. A., Rutz, J. J., Leung, L. Y., Ralph, F. M., Wehner, M., Kawzenuk, B., et al. (2018).

Atmospheric River Tracking Method Intercomparison Project (ARTMIP): project goals

and experimental design. *Geoscientific Model Development*, *11*(6), 2455–2474.

https://doi.org/10.5194/gmd-11-2455-2018

Waliser, D., & Guan, B. (2017). Extreme winds and precipitation during landfall of Atmospheric

rivers. *Nature Geoscience*, *10*(3), 179–183. https://doi.org/10.1038/ngeo2894

Wille, J. D., Favier, V., Dufour, A., Gorodetskaya, I. V., Turner, J., Agosta, C., & Codron, F. (2019).

West Antarctic surface melt triggered by Atmospheric rivers. *Nature Geoscience*, *12*(11),

911–916. https://doi.org/10.1038/s41561-019-0460-1

Xu, G., Ma, X., Chang, P., & Wang, L. (2020). Image-processing-based Atmospheric river tracking

method version 1 (IPART-1). *Geoscientific Model Development*, *13*(10), 4639–4662.

https://doi.org/10.5194/gmd-13-4639-2020